# Learning Structured Universe Graph with Outlier OOD Detection for Partial Matching

**Zetian Jiang**[1†], **Jiaxin Lu**[3†], **Haozhao Fan**[1], **Tianzhe Wang**[1], **Junchi Yan**[12*]

[1]Sch. of Computer Science & Sch. of Artificial Intelligence, Shanghai Jiao Tong University
[2]Shanghai Artificial Intelligence Laboratory
[3]Department of Computer Science, University of Texas at Austin
{maple_jzt,usedtobe,yanjunchi}@sjtu.edu.cn, lujiaxin@utexas.edu

## Abstract

Partial matching is a kind of graph matching where only part of two graphs can be aligned. This problem is particularly important in computer vision applications, where challenges like point occlusion or annotation errors often occur when labeling key points. Previous work has often conflated point occlusion and annotation errors, despite their distinct underlying causes. We propose two components to address these challenges: (1) a structured universe graph is learned to connect two input graphs $\mathbf{X}_{ij} = \mathbf{X}_{iu}\mathbf{X}_{ju}^{\top}$, effectively resolving the issue of point occlusion; (2) an energy-based out-of-distribution detection is designed to remove annotation errors from the input graphs before matching. We evaluated our method on the Pascal VOC and Willow Object datasets, focusing on scenarios involving point occlusion and random outliers. The experimental results demonstrate that our approach consistently outperforms state-of-the-art methods across all tested scenarios, highlighting the accuracy and robustness of our method.

## 1 Introduction

Graph matching has been a fundamental task in traditional computer vision for a long time, applied in areas such as structure from motion (Vijayanarasimhan et al., 2017), object tracking (Nam & Han, 2016; Iqbal et al., 2017), optical flow (Sun et al., 2018), stereo matching (Luo et al., 2016; Chang & Chen, 2018), and pose estimation (Gasse et al., 2019). With the rise of deep learning, integrating graph matching and neural networks has led to significant improvements in accuracy and efficiency (Zanfir & Sminchisescu, 2018; Wang et al., 2019), making it more suitable for downstream tasks. Beyond these, graph matching has also shown great potential in other fields, such as bioinformatics for protein structure comparison (Luo et al., 2021) and drug design (Kriege et al., 2019), as well as in social network analysis for community detection and network alignment (Chen et al., 2020).

In particular, graph matching aims to find node correspondence between graph-structured data via both node and edge affinity (Gold & Rangarajan, 1996; Cho et al., 2010). It is a well-known NP-Hard problem and is often formulated as the quadratic assignment problem (QAP):

$$\max_{\mathbf{X}} \ \text{vec}(\mathbf{X})^{\top}\mathbf{K}\text{vec}(\mathbf{X})$$
$$s.t. \quad \mathbf{X} \in \{0,1\}^{n \times n}, \mathbf{X}\mathbf{1}_n = \mathbf{1}_n, \mathbf{X}^{\top}\mathbf{1}_n = \mathbf{1}_n. \tag{1}$$

Here, $\mathbf{X}$ represents a binary permutation matrix, where $\mathbf{X}_{ij} = 1$ indicates that node $i$ in the first graph corresponds to node $j$ in the second graph. $\mathbf{K} \in \mathbf{R}^{n^2 \times n^2}$ is a hand-crafted or learned affinity matrix whose diagonal (off-diagonal) encodes node (edge) affinities.

While theoretically elegant, the QAP's assumption of perfect one-to-one node correspondence between graphs rarely holds in real-world applications. Practical scenarios often involve partial matching, where only subsets of nodes between graphs correspond, as illustrated in Figure 1. Traditional approaches attempted to address this limitation by introducing dummy nodes with zero affinity. However, this solution proves inadequate for handling real-world challenges such as occlusions and annotation errors.

---
*Correspondence author. † denotes equal contribution. The SJTU authors were in part supported by NSFC (62222607, 92370201), and SJTU Global Strategic Partnership Fund (2023 SJTU-UCL).

Figure 1: Overview of the problem setting and **UGM**. **Partial Matching:** Due to occlusion, viewpoint differences, and annotation errors, only a subset of key points on each graph can be matched. **Left:** Prior works have limited capability in detecting and filtering outliers. **Right: UGM** employs two key components to improve partial matching performance: (1) a universe graph that serves as a bridge to aggregate matches from different graphs, and (2) an outlier filter to remove spurious matches. This design ensures that only inliers are matched, significantly enhancing partial matching performance.

Several recent works have explored ways to overcome this limitation. BBGM (Rolínek et al., 2020a), for instance, bypasses the constraint that each node must find a specific match by learning a negative affinity score. However, this approach heavily relies on the neural network's capacity, and the actual performance gains are often limited. GCAN (Jiang et al., 2022) reformulates graph matching as an integer linear programming (ILP), introducing a placeholder node for each graph to absorb outliers. While this helps, learning for ILP remains a challenging task. AFAT (Wang et al., 2023) predicts which node match to discard by analyzing matched points and the affinity matrix. However, as a post-processing step, its effectiveness is limited because outliers may have already disrupted the initial matching, weakening its ability to correct errors. Another approach, URL (Nurlanov et al., 2023), learns a universe point representation, which serves as a central reference for key point alignment across all images. Although it addresses the issue of point occlusion, it still suffers from redundant matches as the erroneous annotation outliers fail to find the corresponding universe point representation.

A key insight often overlooked by these literature is the distinction between point occlusion and annotation errors: occlusion-caused outliers may find correspondences in other graphs, while annotation errors are inherently meaningless. Building on this observation, we propose **U**niverse **G**raph **M**atching (**UGM**), which addresses these challenges through two key innovations: (1) A latent universe graph learning approach that utilizes the structured universe graph to serve as a bridge for matching graph pairs $\mathbf{X}_{ij} = \mathbf{X}_{iu}\mathbf{X}_{ju}^\top$. This method leverages the inherent structural information from input graphs, enhancing the accuracy and robustness of the learned universe graph for handling point occlusions. (2) An energy-based out-of-distribution detection method that effectively identifies and removes annotation errors at test time. Through training with a margin loss, we effectively differentiate between the energy scores of erroneous annotation outliers and valid in-distribution keypoints. By removing annotation errors, we significantly reduce the complexity of the partial matching problem (as we will discuss in Sec. 3.3), thereby improving overall matching accuracy.

Our primary contributions are:

- We propose Latent Universe Graph Learning, which enhances universe matching robustness through combined node features and structural information.
- We are the first to introduce the Energy-based OOD Detection for annotation error identification, enabling pre-matching outlier filtering for improved accuracy.
- We evaluate the proposed **UGM** approach on keypoint matching tasks under occlusion and random outlier settings using both Pascal VOC and Willow datasets. Our experimental results demonstrate that **UGM** consistently outperforms state-of-the-art methods, highlighting the effectiveness and robustness of our method.

## 2 RELATED WORKS

### 2.1 CLASSICAL GRAPH MATCHING

Two-graph matching has been widely studied, which involves constructing graphs for the objects to be matched and aligning the nodes and edges between the two graphs. (Gold & Rangarajan, 1996) solves graph matching via a graduated assignment (GA) algorithm, which computes the partial derivative

of the objective function with Taylor expansion to turn the GM problem into a linear assignment problem. RRWM (Cho et al., 2010) constructs an association graph for matching graphs and applies the random walk algorithm with matching constraints. RRWHM (Lee et al., 2011) is the extended version of RRWM where the association graph is built with higher-order geometric information extracted from the graph. DS++, DS* (Dym et al., 2017; Bernard et al., 2018) design different ways to tightly relax GM to a convex problem and then project the results back to the original solution space. LPMP (Swoboda et al., 2017) proposes several additional Lagrangian relaxations of the graph matching problem and o leading solvers for this problem optimize the Lagrange decomposition duals with sub-gradient and dual ascent updates.

In addition, numerous works have focused on matching multiple graphs, which is called Multi-Graph Matching. There are mainly two forms of methods. The first solves the problem in discrete space, which includes composition-based method CAO (Yan et al., 2015b), MGM-Floyd (Jiang et al., 2021), tree-structure-based approach MatchOpt (Yan et al., 2015a), DPMC (Wang et al., 2020b), and Majorize-Minimization framework based approach M3C (Lu et al., 2024). Another line of works (Pachauri et al., 2013; Huang & Guibas, 2013; Chen et al., 2014; Wang et al., 2018) etc. tries to relax the MGM problem into a continuous space and re-project the solution to obtain a discrete matching result.

## 2.2 Deep Learning of Graph Matching

Deep learning has recently been applied to graph matching on images (Zanfir & Sminchisescu, 2018), whereby CNN extracts node features from images followed by spectral matching and is learned using a regression-like node correspondence supervision. This work is improved via introducing GNN to encode structural (Wang et al., 2019; Wang et al., 2020) or geometric (Zhang & Lee, 2019) information, with a combinatorial loss based on cross-entropy loss, and Sinkhorn network (Adams & Zemel, 2011) as a differential matching solver. The work (Yu et al., 2020) extends PCA (Permutation loss and Cross-graph Affinity GM) (Wang et al., 2019) by edge embedding and Hungarian-based (Kuhn, 1955) attention mechanism to stabilize end-to-end training. BBGM (Blackbox Deep Graph Matching) (Rolínek et al., 2020b) proposes a better front-end feature extraction backbone with Spline Convolution (Fey et al., 2018), and the gradient is backpropagated by a fitting linear gradient of the discrete graph matching solver (Pogančić et al., 2019). NGM (Neural Graph Matching Nets) (Wang et al., 2021; 2020a) proposes to address the most general Lawler's QAP form, based on the novel feature extractors e.g. (Rolínek et al., 2020a) with the proposed learnable graph matching solver. DLGM (Deep Latent Graph Matching) (Yu et al., 2021), based on BBGM, predicts consistent graph topology utilizing deterministic and generative models to improve the matching quality. GCAN (Jiang et al., 2022) formulates the graph matching problem as an Integer Linear Programming problem and explores a novel node feature extraction framework with attention. AFAT (Wang et al., 2023) uses the attention model to predict which node match to discard by analyzing matched points and the affinity matrix. COMMON (Lin et al., 2023) adopts a contrastive learning approach, utilizing the infoNEC loss to supervise training similarity matrices for both edges and nodes. URL (Nurlanov et al., 2023) introduces a novel universe point representation learning method, applying universe key point matching to the graph matching field for the first time.

## 2.3 Out-of-distribution Detection

Out-of-distribution (OOD) detection is a growing field aimed at identifying data points that deviate from the distribution on which a model was trained. This is crucial for improving the robustness and safety of machine learning systems. Many OOD detection methods can be applied to pre-trained models without modifying the training process. These methods often rely on statistics from the model's output, such as maximum softmax probability, to distinguish between in-distribution (ID) and OOD data (Hendrycks & Gimpel, 2016; Liang et al., 2017). Energy-based OOD detection methods, on the other hand, assign an energy score to each input, where OOD data typically results in higher energy values (Liu et al., 2020; Bao et al., 2024). Generative models, such as VAEs and GANs, have also been used to estimate the likelihood that a given data point belongs to the training distribution (Nalisnick et al., 2018). More recently, self-supervised tasks have been employed to improve OOD detection by learning representations that are highly correlated with the training distribution, enabling the model to identify deviations (Sehwag et al., 2021). Contrastive learning has further advanced OOD detection by ensuring that OOD data points are positioned far from ID data in the learned representation space (Winkens et al., 2020).

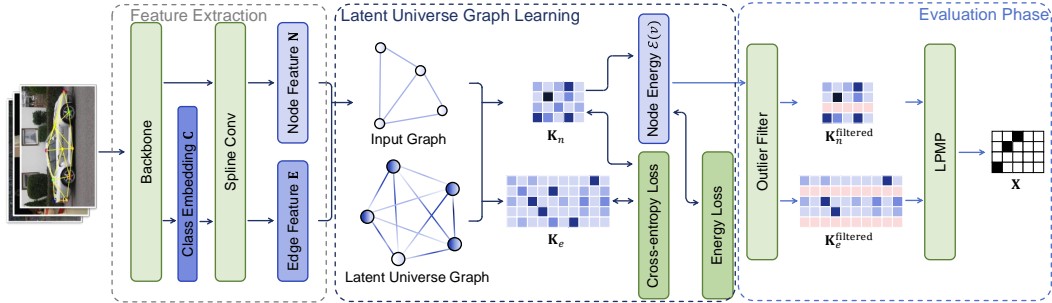

Figure 2: Pipeline of Universe Graph Matching. The pipeline consists of two main components: 1) structured universe graph learning, which includes the training of node and edge affinities, and 2) outlier ood detection through energy score. The dark blue arrows indicate the training phase, while the light blue arrows represent the inference phase.

## 3 METHODOLOGY

In this section, we will introduce our method **U**niverse **G**raph **M**atching (**UGM**) from two parts: structured universe graph learning and outlier ood detection in Section 3.1 and 3.2, respectively. The overall pipeline is shown in Figure 2, and more implementation and training details are provided in Appendix B. Meanwhile, we also discuss **UGM** in perspective of problem formulation and analyze its strength as well as drawbacks in Section 3.3.

### 3.1 STRUCTURED UNIVERSE GRAPH LEARNING

We begin by defining the structured universe graph, which serves as the foundation for our approach. The universe graph $\mathbf{U} = \{\mathbf{N}_u \in \mathbf{R}^{n_u \times d}, \mathbf{E}_u \in \mathbf{R}^{m_u \times d}\}$ is composed of both node embeddings $\mathbf{N}_u$ and edge embeddings $\mathbf{E}_u$. The number of nodes $n_u$ corresponds to the distinct types of key points present in the input graphs. We define this latent $\mathbf{U}$ as a complete, directed graph, where every pair of nodes is connected by a directed edge, resulting in $m_u = n_u \times (n_u - 1)$ edges. This ensures that any edge in the input graph will have a corresponding edge in the universe graph. Moreover, the directed edges capture the inherent asymmetry in real-world relationships between key points, such as varying geometric or contextual dependencies, offering greater flexibility in encoding complex structural information.

We now introduce the pipeline to match the latent universe graph $\mathbf{U}$ to the input graph $\mathbf{G}_i$. The input graph $\mathbf{G}_i = \{\mathbf{N}_i \in \mathbf{R}^{n_i \times d}, \mathbf{E}_i \in \mathbf{R}^{m_i \times d}\}$ consists of node features $\mathbf{N}_i$ and edge features $\mathbf{E}_i$. As in previous works, these features are initially extracted using a CNN-based backbone, followed by refinement through a GNN. Node affinity $\mathbf{K}_n$ and Edge affinity $\mathbf{K}_e$ are constructed by:

$$\mathbf{K}_n = \mathbf{N}_i\mathbf{N}_u^\top, \ \mathbf{K}_e = \mathbf{E}_i\mathbf{E}_u^\top. \tag{2}$$

The matching between the universe graph $\mathbf{U}$ and the input graph $\mathbf{G}_i$ is then obtained using a traditional graph matching solver, such as LPMP (Swoboda et al., 2017):

$$\mathbf{X}_{iu} = GMSolver(\mathbf{K}_n, \mathbf{K}_e). \tag{3}$$

In our training process, we use the ground truth universe matching $\mathbf{X}_{iu}^{gt}$, instead of ground truth pairwise matching $\mathbf{X}_{ij}^{gt}$, as the supervision signal. $\mathbf{X}_{iu}^{gt} \in \{0, 1\}^{n_i \times n_u}$ is a permutation matrix which denotes the ground truth correspondence between input graph $\mathbf{G}_i$ and universe latent graph. In fact, the universe matching set $\{\mathbf{X}_{iu}|1 \le i \le N\}$ and the pairwise matching set $\{\mathbf{X}_{ij}|1 \le i, j \le N\}$ are fully equivalent and can be obtained via non-negative factorisation (Bernard et al., 2019):

$$\mathcal{X}_p = \begin{pmatrix} \mathbf{X}_{00} & \mathbf{X}_{01} & \cdots & \mathbf{X}_{0N} \\ \mathbf{X}_{10} & \mathbf{X}_{11} & \cdots & \mathbf{X}_{1N} \\ \vdots & \vdots & \ddots & \vdots \\ \mathbf{X}_{N0} & \mathbf{X}_{N1} & \cdots & \mathbf{X}_{NN} \end{pmatrix} = \begin{pmatrix} \mathbf{X}_{0u} \\ \mathbf{X}_{1u} \\ \vdots \\ \mathbf{X}_{Nu} \end{pmatrix} \begin{pmatrix} \mathbf{X}_{0u}^\top & \mathbf{X}_{1u}^\top & \cdots & \mathbf{X}_{Nu}^\top \end{pmatrix} = \mathcal{X}_u\mathcal{X}_u^T. \tag{4}$$

Even if some pairwise matchings are missing, we can still recover the universe matching by first applying spectral decomposition and then projecting the result back into the discrete space (Pachauri et al., 2013). It is worth noting that, in most computer vision tasks, the matching matrix is typically

generated from keypoint annotations, and these annotations naturally serve as the ground truth universe matching.

In practice, we cannot directly supervise $\mathbf{X}_{iu}$ with $\mathbf{X}_{iu}^{gt}$, as most graph matching algorithms are non-differentiable. Even when using differentiable algorithms, such as RRWM (Cho et al., 2010), the high number of iterations would make gradient-based optimization difficult. Therefore, instead of supervising the solution, we directly act on the node and edge affinities. By treating the universe's nodes and edges indices as labels, we apply the cross-entropy loss to the node affinity $\mathbf{K}_n$ and edge affinity $\mathbf{K}_e$ for training:

$$\mathcal{L}_{\text{graph}} = CE(\text{softmax}(\mathbf{K}_n), y_n) + CE(\text{softmax}(\mathbf{K}_e), y_e) \tag{5}$$

where

$$y_n = \arg\max(\mathbf{X}_{iu}^{gt}), \; y_e = y_n(EdgeID(0)) \times n_u + y_n(EdgeID(1)). \tag{6}$$

Note $EdgeID \in \mathbb{N}^{2 \times m}$ denotes directed edges in the graph. $EdgeID(0)$ denotes the indices of the source nodes and $EdgeID(1)$ denotes the indices of the target nodes. Another important consideration in the matching process is the diversity of categories. The input may consist of pairs from different categories, each ideally requiring its own distinct latent universe graph. A straightforward approach would be to learn a separate graph representation for each category. However, this method would cause the latent universe graph to grow excessively as the number of categories increases. Consequently, the complexity of the graph matching solver could escalate quadratically or even worse, making it impractical for large-scale problems.

To address this, we propose learning a single latent universe graph, where key points from different categories are mapped to the same universe node. For example, the left front wheel of a car and the left handlebar of a bicycle can share the same universe node embedding. In fact, a single universe node embedding is shared among all categories, allowing the graph's size to be limited by the maximum number of points across all categories. The same principle applies to edge embeddings, which further reduces space complexity.

However, since the latent universe graph shares node and edge embeddings for all categories, the embeddings $\mathbf{N}_u$ and $\mathbf{E}_u$ inherently lack category-specific information. To reintroduce this information for affinity construction, we add a class embedding $\mathbf{C}$ to each key point feature $\mathbf{F}$ extracted by CNN-based backbone. This ensures that the nodes $\mathbf{N}_i$ and edges $\mathbf{E}_i$ of the input graph still carry the necessary class information. The class embedding is derived from the global features $\mathbf{H}$ of the input images pair as follows:

$$\mathbf{C} = Encoder(\mathbf{H}_i, \mathbf{H}_j), \; \mathbf{F}_i = Concat(\mathbf{F}_i, \mathbf{C}), \; \mathbf{F}_j = Concat(\mathbf{F}_j, \mathbf{C}). \tag{7}$$

## 3.2 Outlier Detection

Although the structured universe graph effectively handles point occlusion, it cannot address error annotation outliers, as these outliers do not have corresponding nodes in the universe graph. We frame the identification of these erroneous annotation outliers as an out-of-distribution (OOD) detection problem.

Out-of-distribution detection typically refers to a model's ability to identify and appropriately handle data that deviates significantly from its training distribution. In the context of keypoint annotation, the correctly annotated keypoints adhere to a well-defined, coherent distribution that is consistent across all data. In contrast, erroneously annotated outliers are generated through random and unpredictable factors, causing them to deviate substantially from this established distribution. This inherent difference in data characteristics enables us to frame the outlier detection problem within an OOD detection framework, leveraging the disparity between the structured nature of valid annotations and the stochastic nature of annotation errors.

Following Liu et al. (2020), we define an energy score function for each node $v_i$ of input graph:

$$\mathcal{E}(v_i) = -T \cdot \ln(\sum_j e^{K_n(i,j)/T}), \tag{8}$$

where $K_n(i,j)$ denotes the node affinity between $v_i$ on input graph and $v_j$ on universe graph, and $T$ is a temperature constant. When $v_i$ is an inlier, it should have a corresponding point in the latent universe graph, resulting in a relatively low energy score. Conversely, when $v_i$ is an outlier, all the values of $K_n(i,j)$ will be relatively low, leading to a higher energy score.

While the energy score can be used for a pre-trained neural network directly, the energy gap between in-distribution and out-of-distribution data may not always be optimal for differentiation. To address this, we also introduce a margin loss that fine-tunes the neural network to explicitly create a larger energy gap:

$$\mathcal{L}_{\text{energy}} = \mathbb{E}_{v \sim \mathcal{D}_{in}}(\max(0, \mathcal{E}(v) - m_{in})) + \mathbb{E}_{v \sim \mathcal{D}_{out}}(\max(0, m_{out} - \mathcal{E}(v))), \tag{9}$$

where $\mathcal{D}_{in}$ is the in-distribution training data, $\mathcal{D}_{out}$ is the out-of-distribution training data, $m_{in}$ and $m_{out}$ are the margins for in- and out-of-distribution data, respectively. This loss assigns lower energy scores to in-distribution data and higher scores to OOD data. By contrastively shaping the energy landscape during training, the model becomes more effective at distinguishing between in- and out-of-distribution data.

To apply the energy score in our network, we first compute the energy $\mathcal{E}$ for each node in the input graph $\mathbf{G}_i$ based on its affinity with nodes in the latent universe graph $\mathbf{U}$. Nodes with energy scores above a certain threshold $\tau$, indicating a high likelihood of being outliers, are removed before the graph matching process:

$$\mathbf{G}^{\text{filtered}} = \{\mathbf{V} = \{v_i | \mathcal{E}(v_i) \leq \tau\}, \mathbf{E} = \{e_{ij} | \mathcal{E}(v_i) \leq \tau, \mathcal{E}(v_j) \leq \tau\}\} \tag{10}$$

We use the newly obtained graph $\mathbf{G}^{\text{filtered}}$ to construct the filtered node affinity $\mathbf{K}_n^{\text{filtered}}$ and edge affinity $\mathbf{K}_e^{\text{filtered}}$, and then feed it into the graph matching solver to obtain the matching result.

$$\mathbf{X}_{iu}^{\text{filtered}} = GMSolver(\mathbf{K}_n^{\text{filtered}}, \mathbf{K}_e^{\text{filtered}}). \tag{11}$$

This preprocessing step improves the quality and accuracy of the subsequent matching between the input graph and the latent universe graph. After the matching process, the removed outlier points are reintroduced into the solution to maintain the integrity of the input data.

$$\mathbf{X}_{iu}(v_i) = \begin{cases} \mathbf{0}^\top & \mathcal{E}(v_i) > \tau, \\ \mathbf{X}_{iu}^{\text{filtered}}(v_i) & \mathcal{E}(v_i) \leq \tau. \end{cases} \tag{12}$$

## 3.3 Discussion

Universe Graph Matching essentially decomposes the ambiguous partial matching problem into two well-defined subgraph matching problems. Partial matching, where only parts of the input graphs are aligned, is difficult to formalize with clear optimization objectives and constraints. Conventional approaches, such as adding dummy nodes or relaxing the condition $\mathbf{X}\mathbf{1}_n = \mathbf{1}_n$ to $\mathbf{X}\mathbf{1}_n \leq \mathbf{1}_n$, have proven inadequate. Our key contribution lies in introducing latent universe graph, which enables us to decompose the partial matching problem between two graphs as $\mathbf{X}_{ij} = \mathbf{X}_{iu}\mathbf{X}_{ju}^\top$. This reformulation shifts the objective from maximizing the match $\mathbf{X}_{ij}$ between the input graphs to maximizing the match $\mathbf{X}_{iu}, \mathbf{X}_{ju}$ between each input graph and the latent universe graph. Since the latent universe graph contains all points and edges, the input graphs become its subgraphs, turning the solution of $\mathbf{X}_{iu}$ into a well-defined subgraph matching problem. This more precise and structured problem definition in universe graph matching improves performance over traditional pairwise graph matching on partial matching tasks.

However, the latent universe graph alone cannot fully address the error annotation points, which lack corresponding matches in the universe graph and violate the assumption that input graphs are subgraphs of the universe graph. While introducing dummy nodes in to the universe graph $\mathbf{U}$ could potentially account for these outliers, this approach faces several challenges. Estimating the appropriate number of dummy nodes is difficult, and the randomness of error annotations makes it complicate to learning effective dummy node embeddings. Our solution adopts an OOD detection approach, treating error annotation outliers as out-of-distribution points and excluding them before solving the subgraph matching problem.

Despite its advantages, our approach still faces several challenges: 1) Universe latent graph is inherently limited to closed-set problems, relying on class information and training data, which restricts generalization to unseen categories. However, few-shot learning can significantly alleviate this limitation, as shown in our experiments in Appendix **??**, demonstrating the method's potential to adapt to new classes. 2) The universe graph is a complete graph; however, the distribution of edges in the input graphs is uneven. This mismatch leads to suboptimal learning of many edge embeddings, with some edges either absent or appearing only rarely in the training data. 3) While the universe graph is more stable and structured than universe point representation, the universe graph cannot fully prevent errors in $\mathbf{X}_{iu}$ from propagating to all $\mathbf{X}_{ij}$ matches. All in all, expanding graph partial matching to a more general form, both in terms of mathematical modeling and finer design, requires a long-term exploration.

# 4 EXPERIMENT

## 4.1 PROTOCOL

**Datasets**  We evaluate our method on Pascal VOC (Everingham et al., 2010) and Willow Object Class (Cho et al., 2013), two widely recognized datasets.

The Pascal VOC (Everingham et al., 2010) with Berkeley annotations  (Bourdev & Malik, 2009) contains images with bounding boxes surrounding objects of 20 classes. We follow the standard data preparation procedure of NGM and BBGM (Wang et al., 2021; Rolínek et al., 2020b). Each object is cropped to its bounding box and scaled to $256 \times 256$ px. We follow the unfiltered setting (Rolínek et al., 2020b), which preserves all key points for graph pairs, to evaluate our method's performance under point occlusion. Moreover, we randomly sample the coordinates on images to add outliers, which serve as error annotation key points. We also follow prior works to split the train and test dataset, where training data includes 7,020 images and test data includes 1,682 images.

The Willow Object Class (Cho et al., 2013) contains images from Caltech-256 (Griffin et al., 2007) and Pascal VOC 2007 (Everingham et al.), which consists of 256 images from 5 categories: 40 cars, 40 motorbikes, 50 ducks, 66 wine bottles, and 109 faces. Each image is annotated with the same ten key points. Following standard procedure, we also rescale the image to $256 \times 256$ px. In experiments, we randomly drop 0 - 6 key points for each image for the 'occlusion' setting. We also add 1 - 10 random outliers as we do in PascalVOC for the 'random outlier' setting. We choose 20 images from each category as our training dataset and leave others for evaluation.

**Compared Method**  We compare our methods **UGM** with previous work of two kinds: traditional solver with pre-defined affinity and deep graph matching learning model. For the learning-free traditional solvers, we consider four works:  GAGM (Gold & Rangarajan, 1996), RRWM (Cho et al., 2010), BPF (Wang et al., 2017), and ZAC (Wang et al., 2020). We generate node and edge affinities using the pre-trained BBGM learning framework and input these affinities into each solver to obtain their respective matching results. For learning-based models, we compare our method with NGMv2 (Wang et al., 2021), BBGM (Rolínek et al., 2020b), DLGM (Yu et al., 2021), GCAN (Jiang et al., 2022), AFAT (Wang et al., 2023), and URL (Lin et al., 2023). To report the performance of prior works, we adhere to the following principles: 1) If a prior work's experimental settings align with ours, we directly report the performance metrics provided in their paper. 2) If the experimental setting is unique to our work, we attempt to replicate the prior methods under their original experimental settings and then adapt them to our setting to obtain comparable performance metrics. However, we failed to replicate the methods of DLGM and URL because they did not release publicly available code.  Therefore, we did not report their performance in some of the experiments (Table 2 and Table 3).

**Evaluation Metric**  We see the matching prediction as a binary classification task for each entry. Therefore, F1 score is applied as an evaluation metric, which is introduced by BBGM (Rolínek et al., 2020b). The F1 score is calculated as follows:

$$\mathbf{F}_1 = 2 \cdot \frac{\text{Precision} \cdot \text{Recall}}{\text{Precision} + \text{Recall}} = \frac{2\text{TP}}{2\text{TP} + \text{FP} + \text{FN}} \tag{13}$$

where TP denotes true positive, FP denotes false positive, and FN denotes false negative.

## 4.2 PERFORMANCE ON PASCAL VOC

To thoroughly evaluate the performance of our proposed **UGM** method, we conducted experiments on the Pascal VOC dataset under two distinct settings: the unfiltered setting, where all key points are preserved without the introduction of additional noise, and the random outlier setting, where two random outliers are further added to each input image. All the results are shown in Table 1 and  2.

Our proposed **UGM** method demonstrates superior performance in both setting, surpassing the state-of-the-art (SOTA) models by a significant margin. Specifically, in the unfiltered setting, **UGM** outperforms the best-performing SOTA model by 2.2% in terms of average F1 score. Similarly, in the random outlier setting, **UGM** still leads the comparison, exceeding the next best method by 4.8%. This consistent improvement across both settings highlights the robustness and generalization ability of our approach.

A comparison of the two tables reveals how much performance degrades between the unfiltered and random outlier settings, allowing us to evaluate the robustness of learning-based models in the presence of outliers. Our UGM method experiences a performance drop of 8.8%, ranking second

Table 1: F1 Score performance on Pascal VOC under 'unfiltered' setting. All the key points are preserved, and no additional outliers are added. **Top**: learning-free graph matching solver with pre-learned affinity. **Middle**: deep learning graph matching model. **Bottom**: out method **UGM**.

| Method | aero | bike | bird | boat | bottle | bus | car | cat | chair | cow | table | dog | horse | motor | person | plant | sheep | sofa | train | tv | avg |
|---|---|---|---|---|---|---|---|---|---|---|---|---|---|---|---|---|---|---|---|---|---|
| GAGM | 37.3 | 60.3 | 47.7 | 39.1 | 75.0 | 53.7 | 36.1 | 59.2 | 34.8 | 56.2 | 40.2 | 54.4 | 51.9 | 54.3 | 39 | 83.8 | 48.8 | 19 | 58.3 | 68.7 | 50.9 |
| RRWM | 39.4 | 63.7 | 52.1 | 39.7 | 76.5 | 57.9 | 28.5 | 64.3 | 37.9 | 60.3 | 43 | 57.9 | 54.2 | 54 | 43.3 | 85 | 50.4 | 22 | 66.1 | 69.9 | 53.3 |
| BPF | 39.2 | 66.5 | 51.2 | 40.0 | 77.0 | 59.0 | 26.7 | 63.8 | 36.9 | 60.6 | 49.7 | 58.1 | 54.3 | 58.2 | 43.6 | 84.9 | 50.7 | 23 | 66.4 | 70.4 | 54.0 |
| ZAC | 43.5 | 62.8 | 57.5 | 43.9 | 72.1 | 61.2 | 33.6 | 69 | 38.3 | 64.1 | 46.4 | 66.2 | 61.8 | 61.4 | 46.4 | 83.2 | 58.1 | 27.9 | 69.8 | 72.6 | 57.0 |
| NGMv2 | 45.5 | 65.3 | 55.3 | 45.8 | 88.4 | 64.3 | 45.9 | 58.6 | 43.3 | 59.1 | 39.2 | 55.7 | 58 | 65.3 | 44.4 | 95.4 | 50.3 | 41.2 | 72.4 | 81.8 | 58.8 |
| BBGM | 42.7 | 70.9 | 57.5 | 46.6 | 85.8 | 64.1 | 51 | 63.8 | 42.4 | 63.7 | 47.9 | 61.5 | 63.4 | 69 | 46.1 | 94.2 | 57.4 | 39 | 78 | 82.7 | 61.4 |
| DLGM | 43.8 | 72.9 | 58.5 | 47.4 | 86.4 | 71.2 | 53.1 | 66.9 | 54.6 | 67.8 | 64.9 | 65.7 | 66.9 | 70.8 | 47.4 | 96.5 | 61.4 | 48.4 | 77.5 | 83.9 | 64.8 |
| GCAN | 45 | 66.7 | 60.6 | 49.7 | 89.7 | 66.3 | 65.2 | 64.9 | 45.5 | 66.9 | 54.4 | 63.1 | 62.5 | 63.5 | 55 | 96.1 | 63.5 | 49.7 | 80.6 | 83.6 | 64.6 |
| AFAT | 47.1 | 70.8 | 58.1 | 45.8 | 90.8 | 66.5 | 49.6 | 58.8 | 50.6 | 64.6 | 47.2 | 60.5 | 62.3 | 65.7 | 46.3 | 95.4 | 52.7 | 47.4 | 74.2 | 83.8 | 62.0 |
| URL | **62.7** | 75.2 | **73** | **56.7** | **93.7** | 66.2 | 76.7 | 69.2 | 64.9 | **76.6** | 44.6 | 74.4 | **78.8** | 80.9 | 62.5 | 96.9 | 70.3 | 55.4 | 73.6 | 82.1 | 71.7 |
| UGM | 57.4 | **76.3** | 71.2 | 55.1 | 91.7 | **72.3** | **80.3** | **72.3** | **70.1** | 74.6 | **61.5** | **75.3** | 71.6 | 76.1 | **63.8** | **97.2** | **73.6** | **60.0** | **93.1** | **84.8** | **73.9** |

Table 2: F1 Score performance on Pascal VOC under 'random outlier' setting. Two random outliers are further added to each input image. **Top**: learning-free graph matching solver with pre-learned affinity. **Middle**: deep learning graph matching model. **Bottom**: out method **UGM**.

| Method | aero | bike | bird | boat | bottle | bus | car | cat | chair | cow | table | dog | horse | motor | person | plant | sheep | sofa | train | tv | avg |
|---|---|---|---|---|---|---|---|---|---|---|---|---|---|---|---|---|---|---|---|---|---|
| GAGM | 26.7 | 49.3 | 36.6 | 25.1 | 61.9 | 40.7 | 9.0 | 45.4 | 22.5 | 39.3 | 13.1 | 41.0 | 37.8 | 39.1 | 27.9 | 59.8 | 29.1 | 19.7 | 26.0 | 54.9 | 35.2 |
| RRWM | 28.3 | 54.6 | 40.3 | 27.4 | 70.6 | 46.3 | 9.0 | 49.8 | 23.5 | 42.8 | 15.1 | 44.8 | 38.1 | 39.1 | 30.3 | 66.8 | 29.3 | 18.1 | 27.1 | 64 | 38.3 |
| BPF | 25.8 | 55.2 | 38.8 | 26.0 | 70.7 | 45.7 | 9.5 | 49.4 | 22.9 | 40.2 | 14.4 | 44.1 | 36.9 | 40.0 | 29.0 | 68.6 | 28.6 | 18.8 | 28.1 | 63.8 | 37.8 |
| ZAC | 37.2 | 62.3 | 50.8 | 34.0 | 76.0 | 55.9 | 23.8 | 63.7 | 33.2 | 58.8 | 37.7 | 59.4 | 55.8 | 55.4 | 39.1 | 76.8 | 49.3 | 29.0 | 49.1 | 68.9 | 50.8 |
| NGMv2 | 35.6 | 60.2 | 45.0 | 34.3 | 72.1 | 51.3 | 40.2 | 50.2 | 29.5 | 49.4 | 23.7 | 47.6 | 49.8 | 51.5 | 38.7 | 69.3 | 42.9 | 40.8 | 45.0 | 62.2 | 47.0 |
| BBGM | 34.4 | 63.9 | 44.1 | 35.0 | 78.5 | 57.6 | 21.5 | 54.6 | 33.7 | 50.3 | 37.4 | 49.8 | 50.8 | 54.7 | 29.6 | 79.8 | 44.1 | 23.5 | 44.5 | 76.1 | 48.2 |
| GCAN | 44.4 | 61.7 | 51.2 | 47.5 | 78.8 | **72.3** | 48.9 | 58.0 | 50.2 | 62.8 | **53.0** | 54.6 | 60.3 | 61.2 | 51.5 | 83.2 | **62.6** | **48.8** | **70.0** | **84.8** | 60.3 |
| AFAT | 41.2 | 66.9 | 51.4 | 36.5 | 82.7 | 61.8 | 37.9 | 56.9 | 34.8 | 55.7 | 28.0 | 54.4 | 55.3 | 60.3 | 38.2 | 86.3 | 48.8 | 32.6 | 57.1 | 77.1 | 53.1 |
| UGM | **49.9** | **72.2** | **63.5** | **48.8** | **85.3** | 70.1 | **71.8** | **66.5** | **54.6** | **67.4** | 49.5 | **67.4** | **65.2** | **66.2** | **57.6** | **88.0** | 62.2 | 47.2 | 65.6 | 83.4 | **65.1** |

among the learning-based models, just behind GCAN. In comparison, models like NGMv2 and BBGM exhibit larger performance declines of 11.8% and 13.2%, respectively. The relatively low decrease in **UGM**'s performance demonstrates its robustness and adaptability, particularly in noisy and complex environments, making it a reliable option for real-world graph matching tasks.

We also perform an random outlier pressure test on our method **UGM**. Results are shown in Figure 3. Classes like sofa, train, and table see a significant performance drop (32% - 61%) as the number of random outliers increases. This decline is primarily due to the limited amount of training data, particularly for classes like 'table', which has only 28 images, and 'sofa', which has 73 images in the train dataset. The fewer images available, the lower the quality of the constructed latent universe graph, making it more difficult to distinguish between inliers and error annotation outliers. On the other hand, classes like TV, boat, horse, and bus are more robust, with relatively smaller drops in performance (9.9% - 13.9%).

## 4.3 PERFORMANCE ON WILLOW OBJECT

In this experiment, we evaluate the performance of various learning-based graph matching models on the Willow dataset under three settings: occlusion, random outlier, and occlusion + random outlier. These settings simulate real-world challenges, where key points may be missing (occlusion), or noisy data (outliers) may be introduced. All the results are shown in Table 3.

Our proposed **UGM** method demonstrates superior performance across all three settings. In the occlusion setting, **UGM** achieves the highest average F1 score of 91.9%, outperforming the next best model, GCAN, by a significant margin of 9.7%. In the random outlier setting, **UGM** again leads with an average score of 85.4%, surpassing AFAT, the second-best model, by 1.7%. Finally, in the occlusion + random outlier setting, **UGM** shows exceptional robustness, maintaining the highest average score of 71.1%, which is 5.9% higher than the second-place model.

It is worth noting that the extent of our lead varies across the three settings.

- In the occlusion setting, **UGM** demonstrates a significant advantage. This suggests that the latent universe graph structure offers a considerable benefit compared to pairwise matching approaches,

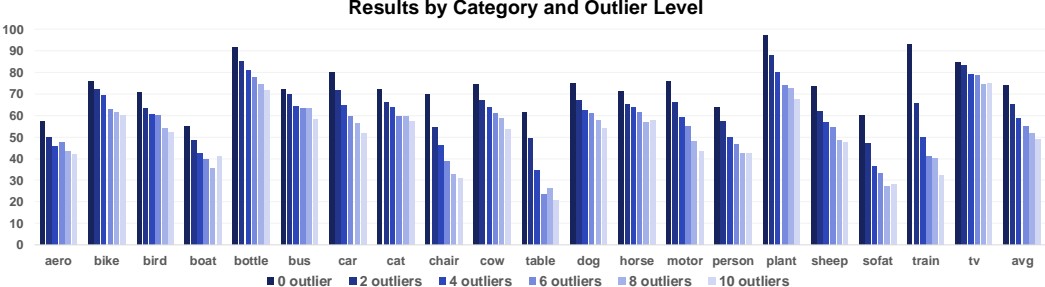

Figure 3: Random outlier pressure test of **UGM** on Pascal VOC. All the key points are preserved, and additional outliers are added to each input image.

Table 3: F1 Score performance on Willow Object Class. **Occlusion**: randomly drop 0-6 key points for each input image. **Random Outlier**: randomly add 1-10 outliers for each input image. **Occlusion + Random Outlier**: both operations are applied. Only the learning model's performance is reported.

| Method | Occlusion | | | | | | Random Outlier | | | | | | Occlusion + Random Outlier | | | | | |
|--------|------|------|------|------|--------|------|------|------|------|------|--------|------|------|------|------|------|--------|------|
| | car | duck | face | motor | bottle | avg | car | duck | face | motor | bottle | avg | car | duck | face | motor | bottle | avg |
| NGMv2 | 80.9 | 73.8 | 84.9 | 85.7 | 76.7 | 80.4 | 78.9 | 66.6 | 84.3 | 63.1 | 76 | 73.8 | 46.3 | 45.2 | 61.6 | 42.9 | 54.6 | 50.1 |
| BBGM | 77.5 | 78.8 | 92.7 | 83.1 | 82.3 | 82.9 | 65.1 | 60.7 | 85.5 | 71.6 | 65.5 | 69.7 | 36.2 | 35.5 | 57.2 | 35.0 | 52.4 | 43.3 |
| GCAN | 81.4 | 74.4 | 88.5 | 88.3 | 78.2 | 82.2 | 74.8 | 75.7 | 92.8 | 77.1 | 83.5 | 80.8 | 43.7 | 51.7 | 60.2 | 45.8 | 56.9 | 51.7 |
| AFAT | 77.1 | 73.7 | 88.0 | 76.6 | 86.4 | 80.4 | 82.2 | **77.7** | 92.7 | 77.2 | **88.6** | 83.7 | **70.8** | 54.7 | 79.1 | 53.8 | 67.9 | 65.2 |
| UGM | **87.7** | **86.2** | **99.5** | **90.2** | **96.0** | **91.9** | **88.1** | 72.9 | **96.8** | **82.5** | 86.5 | **85.4** | 64.4 | **62.1** | **89.6** | **65.1** | **74.5** | **71.1** |

as Universe Graph Matching essentially decomposes the ambiguous pairwise partial matching problem into two well-defined subgraph matching problems, thereby improving its ability to handle missing key points.

- In the random outlier setting, the performance gap between **UGM** and other models is narrower. While our OOD (out-of-distribution) detection technique offers a clear advantage, the improvement is less pronounced. This is likely due to the fact that our approach does not introduce additional training parameters, which limits the potential for further gains.

- Finally, in the occlusion + random outlier setting, **UGM** once again achieves a clear lead. This demonstrates the strong synergy between the latent universe graph learning and our energy based OOD detection technique. The two techniques work in harmony, with no conflicting or contradictory effects, allowing **UGM** to excel in the most challenging conditions.

## 4.4 ABLATION STUDIES

The ablation study results presented in Table 4 highlight the contribution of each key component—class embedding, universe graph, and outlier filter—to the overall performance. Without the outlier filter component, we matched all the points in the input graph to the latent universe graph. Without edge learning, we only learned the universe node representation and used a linear assignment solver to obtain the matching results between the input graph and the universe point representation. Without class embedding, we directly used the local key point features extracted by the backbone to learn the node embeddings.

Table 4: Ablation study on Pascal VOC with 'random outlier' setting. All the key points are preserved, and two random outliers are added to each input image. F1 score is reported.

| class embedding | edge learning | outlier filter | F1 score |
|:---:|:---:|:---:|:---:|
| ✔ | ✔ | ✔ | 65.12 |
| ✔ | ✔ | ✘ | 56.88 |
| ✔ | ✘ | ✔ | 63.49 |
| ✘ | ✔ | ✔ | 58.71 |

These results demonstrate that each component plays a unique role in enhancing the model's performance. Specifically, removing the outlier filter has the most significant negative impact, underscoring its importance in handling noise and maintaining robust graph matching.

Furthermore, we also conduct a sensitivity test for hyperparameters on Pascal VOC under the 'random outlier' setting, as shown in Figure 4. In this sensitivity analysis, we investigate the effects of varying three hyperparameters—temperature $T$, threshold $\tau$, and the pair of values for $m_{in}$ and $m_{out}$. Unless

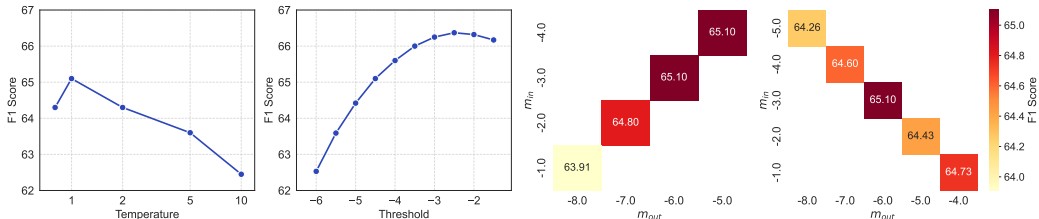

Figure 4: Sensitivity test for hyperparameters on Pascal VOC under 'random outlier' setting. Temperature $T$ in energy function, OOD threshold $\tau$, and $m_{in}, m_{out}$ in energy loss are tested.

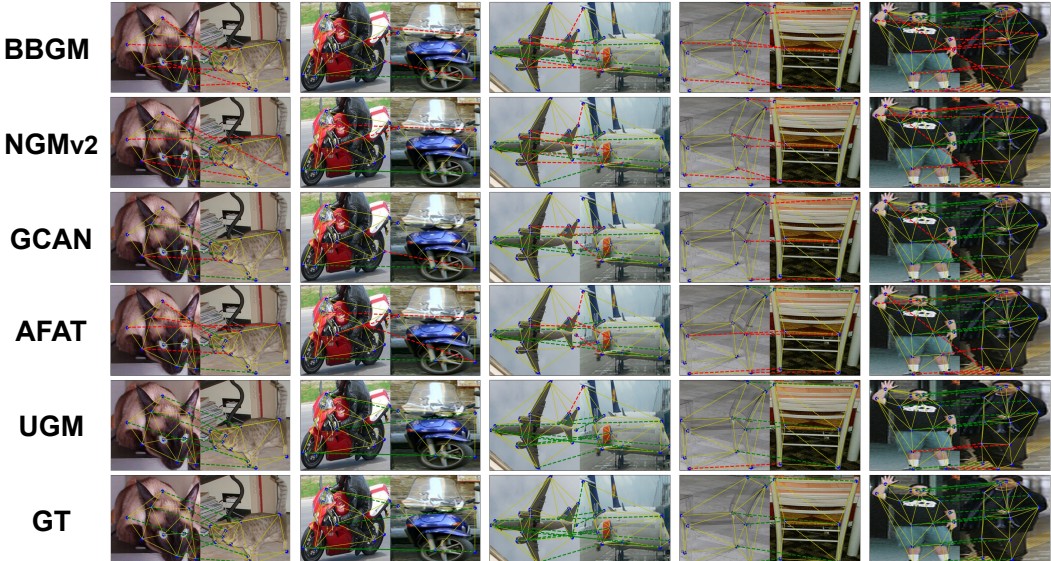

Figure 5: Visualization on Pascal VOC of **UGM** and peer methods. The green dashed lines represent correct matches, while the red dashed lines represent incorrect matches. Ground truth matching is shown at the bottom.

specified otherwise, the default parameter values are set to $T = 1.0$, $m_{in} = -3$, $m_{out} = -6$, and $\tau = (m_{in} + m_{out})/2$.

For the temperature experiment, we tested values ranging from 0.8 to 10. The results show that performance peaks at temp=1.0 with an average value of 65.1, and gradually decreases as the temperature increases. The threshold experiment explores a range from -1.5 to -6. The average performance is highest at $\tau$=-2.5 with a value of 66.37, and the performance decreases consistently as the threshold becomes more negative. Although adjusting $\tau$ can achieve a higher F1 score, we are concerned that this might lead to some overfitting in the model. Therefore, we still report the model results using the default value $\tau = (m_{in} + m_{out})/2$ in other contexts. For the $m_{in}$ and $m_{out}$ experiments, we analyze both their margin (difference) and absolute offset. Performance peaks at (-3, -6), where the margin is 3, but the overall impact of varying the range (from 1 to 7) or shifting their absolute values is minimal.

## 5 CONCLUSION

In summary, this work presents a novel approach to addressing the challenges of partial matching in graph-based keypoint alignment tasks. We introduce Structured Universe Graph Learning to effectively resolve point occlusion by connecting input pairs via a learned latent graph. Additionally, by incorporating both node features and structural information, the robustness of the matching process is further enhanced. We also adopt Energy-based Out-of-Distribution Detection to filter out annotation errors before matching, improving the overall quality of the matching process. Through extensive evaluations of the Pascal VOC and Willow Object datasets, our method consistently outperforms state-of-the-art techniques, particularly in challenging scenarios involving both point occlusion and random outliers, demonstrating the effectiveness and robustness of our approach.

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

# APPENDIX

## A  NOTATION

We first present all notations used in this paper for a better understanding of proposed algorithms and to facilitate the following discussion.

Table 5: Main notations and description used in this paper.

| Notations | Descriptions |
|---|---|
| $\mathbf{N}$ | Node embedding or node feature on the graph. |
| $\mathbf{E}$ | Edge embedding or node feature on the graph. |
| $\mathbf{U}$ | $\mathbf{U} = \{\mathbf{N}_u \in \mathbf{R}^{n_u \times d}, \mathbf{E}_u \in \mathbf{R}^{m_u \times d}\}$ denotes the latent universe graph, where $n_u$ and $m_u$ represents the node and edge number of the latent universe graph. |
| $\mathbf{G}_i$ | $\mathbf{G}_i = \{\mathbf{N}_i \in \mathbf{R}^{n_i \times d}, \mathbf{E}_i \in \mathbf{R}^{m_i \times d}\}$ denotes the input graph, where $n_i$ and $m_i$ represents the node and edge number of the input graph. |
| $EdgeID$ | $EdgeID \in \mathbb{N}^{2 \times m}$ denotes directed edges in the graph. $EdgeID(0)$ denote the indices of the source nodes and $EdgeID(1)$ denote the indices of the target nodes. |
| $\mathbf{K}_n$ | $\mathbf{K}_n \in \mathbf{R}^{n_i \times n_u}$ denotes the node affinity between the input graph and the universe graph. |
| $\mathbf{K}_e$ | $\mathbf{K}_e \in \mathbf{R}^{m_i \times m_u}$ denotes the edge affinity between the input graph and the universe graph. |
| $\mathbf{X}_{ij}$ | $\mathbf{X}_{ij} \in \{0,1\}^{n_i \times n_u}$ denotes the matching between the input graphs. |
| $\mathbf{X}_{iu}$ | $\mathbf{X}_{ij} \in \{0,1\}^{n_i \times n_u}$ denotes the matching between the input graph and the universe graph. |
| $\mathbf{C}$ | Class embedding for image pair. |
| $\mathbf{H}_i$ | Global feature extracted by backbone for the input image. |
| $\mathbf{F}_i$ | Key point feature extracted by backbone for the input image. |
| $\mathcal{E}(v_i)$ | Energy score for node $v_i$. |
| $\tau$ | Energy score threshold for outlier filter. |
| $\mathcal{D}_{in}, \mathcal{D}_{out}$ | $\mathcal{D}_{in}, \mathcal{D}_{out}$ denotes in-distribution data and out-of-distribution data respectively. |

## B  IMPLEMENTATION DETAIL

We utilize the standard feature extractor pipeline with a few modifications.

- Fisrt of all, we replace the VGG16 (Simonyan & Zisserman, 2014) backbone with ResNet50 (He et al., 2016). We compute the outputs of *relu3_5*, *relu4_1* of the ResNet50 network pre-trained on ImageNet (Krizhevsky et al., 2012), to obtain feature $\mathbf{F}_1$ and $\mathbf{F}_2$, respectively. These features are then concatenated to create the key point feature $\mathbf{F}$:

$$\mathbf{F} = CONCAT(\mathbf{F}_1, \mathbf{F}_2) \tag{14}$$

Class embedding is added to $\mathbf{F}$ as introduced in Equation 7, and we apply an MLP layer to reduce the feature dim to keep the same dimension with the output of the VGG16-based backbone.

- Then we feed the obtained feature $\mathbf{F}$ and the graph adjacency $\mathcal{A}$ into the geometric feature refinement component. The graph adjacency $\mathcal{A}$ is generated using Delaunay triangulation (Delaunay et al., 1934) based on keypoint locations. We apply SplineConv (Fey et al., 2018) to encode higher-order information and the geometric structure of the entire graph into node-wise features $\mathbf{F}^n$:

$$\mathbf{N}, \mathbf{E} = SplineConv(\mathbf{F}, \mathcal{A}) \tag{15}$$

The Spline Conv operation is calculated as follows:

$$\mathbf{F}_i = \frac{1}{|\mathcal{N}(i)|} \sum_{j \in \mathcal{N}(i)} \mathbf{F}_j \cdot h_\Theta(\mathbf{F}_i - \mathbf{F}_j) \tag{16}$$

where $\mathbf{F}_i$ represents the node feature of $v_i$, $\mathcal{N}(i)$ denotes the neighborhood of $v_i$, and $h_\Theta$ denotes a kernel function defined over the weighted B-Spline tensor product basis.

---

**Algorithm 1 Universe Graph Matching**

---

**Require:** Input images $I$, input key points $P$, learnable parameters $\theta$, learning rate $\eta$, epoch number $E$, margins $m_{in}, m_{out}$, Temperature $T$, OOD threshold $\tau$
**Ensure:** Trained parameters $\theta^*$ and universe latent graph $\mathbf{U}^*$
 1: Initialize $\theta$ randomly
 2: Initialize node and edge embedding of universe latent graph $\mathbf{U}$ randomly
 3: **for** $e = 1$ to $E$ **do**
 4:     **for** image pair $I_i, I_j$ in train/test dataset **do**
 5:         *# extract features*
 6:         Extract key point feature $\mathbf{F}$ by CNN backbone via Eq. 14
 7:         Add class embedding to $\mathbf{F}$ via Eq. 7
 8:         Refine $\mathbf{F}$ with Spline Conv to obtain node $\mathbf{N}$ and edge feature $\mathbf{E}$ via Eq. 15 and Eq. 16
 9:
10:         *# build affinity*
11:         Construct node affinity $\mathbf{K}_n$ and edge affinity $\mathbf{K}_e$ via Eq. 2
12:         Calculate energy $\mathcal{E}$ for each node via Eq. 8
13:         Filter out random outlier with OOD threshold $\tau$ via Eq. 10
14:
15:         **if** training **then**
16:             *# loss and update*
17:             Calculate permutation loss with $\mathbf{K}_n^{\text{filtered}}$ and $\mathbf{K}_e^{\text{filtered}}$ for both $I_i$ and $I_j$
18:             Calculate energy loss with $\mathcal{E}$, $m_{in}$, and $m_{out}$ via Eq. 9 for both $I_i$ and $I_j$
19:             Final loss $\mathcal{L} = \mathcal{L}_{\text{permutation}} + \mathcal{L}_{\text{energy}}$, and compute gradient $\nabla_\theta \mathcal{L}, \nabla_{\mathbf{U}} \mathcal{L}$
20:             Update parameters: $\theta \leftarrow \theta - \eta \nabla_\theta \mathcal{L}$
21:             Update universe embedding: $\mathbf{U} \leftarrow \mathbf{U} - \eta \nabla_{\mathbf{U}} \mathcal{L}$
22:         **else**
23:             *# build pairwise matching*
24:             Use LPMP solver to obtain universe matching $\mathbf{X}_{iu}, \mathbf{X}_{ju}$ via Eq. 11 and Eq. 12
25:             Build pairwise matching by $\mathbf{X} = \mathbf{X}_{iu}\mathbf{X}_{ju}^\top$
26:         **end if**
27:     **end for**
28: **end for**
29: **return** $\theta^*$, $\mathbf{U}^*$

---

Node and edge affinity construction, universe matching, and outlier filter process have been introduced in Section 3.1 and 3.2. We finally build pairwise matching by $\mathbf{X}_{ij} = \mathbf{X}_{iu}\mathbf{X}_{ju}^\top$.

All the models are trained on on a Linux workstation with i9-10920X CPU@3.50GHz CPU, one RTX3090, and 128GB RAM. The training and inference algorithm is shown in Algorithm 1. In default, we train our **UGM** with hyper parameters $m_{in} = -6$, $m_{out} = -3$, $T = 1.0$, $\tau = -4.5$, $\eta = 1e-3$, and $E = 15$.

## C  BACKBONE COMPARISON

To systematically evaluate the impact of network architecture on performance, we conducted comprehensive experiments comparing VGG and ResNet as backbone models across all methods. This analysis ensures fair comparison by providing results using consistent architectures while also quantifying how architectural choices influence each method's effectiveness.

As demonstrated in Table 6, our method consistently outperforms existing approaches across multiple feature extractor configurations, including VGG+Spline2D, VGG+Attention, and ResNet+Spline2D. The sole exception occurs with VGG+Spline2D+3D, where URL achieves better results through its utilization of spline3D features. However, due to the unavailability of URL's implementation, we were unable to integrate this feature extractor into our UGM model. To facilitate a fair comparison, we adopted ResNet50 as the common backbone architecture.

While evaluating architectural impacts, we observed that not all baseline methods exhibited performance improvements when transitioning to ResNet. To maintain scientific rigor and transparency, we have included these results in this appendix, allowing for a more proper discussion of architectural sensitivities across different approaches.

Table 6: Backbone fairness comparison on Pascal VOC with unfiltered setting. The performance with the VGG backbone is reported by their paper, and the performance with the ResNet backbone is reproduced via publicly available code. '-' denotes that the method has not released its code.

| Method | Backbone | GNN | F1 score | Backbone | GNN | F1 score |
|--------|----------|-----|----------|----------|-----|----------|
| NGMv2 | VGG | Spline2D | 58.8 | ResNet | Spline2D | 57.46 |
| BBGM | VGG | Spline2D | 61.4 | ResNet | Spline2D | 64.45 |
| DLGM | VGG | Spline2D | 64.8 | ResNet | Spline2D | - |
| GCAN | VGG | Attention | 64.6 | ResNet | Attention | 64.67 |
| AFAT | VGG | Attention | 62 | ResNet | Attention | 59.55 |
| URL | VGG | Spline2D | 67.6 | ResNet | Spline2D | - |
| URL | VGG | Spline2D+3D | 71.7 | ResNet | Spline2D+3D | - |
| UGM | VGG | Spline2D | 70.5 | ResNet | Spline2D | 73.9 |

# D   EFFECT OF ENERGY MARGIN LOSS

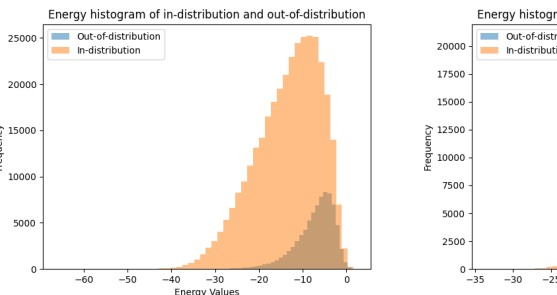

Figure 6: **Left**: Energy histogram of ID and OOD samples without energy loss finetuning, **Right**: Energy histogram of ID and OOD samples with energy loss finetuning

The effect of energy margin loss is shown in Figure 6. Left shows the energy distribution of in-distribution (ID) and out-of-distribution (OOD) samples before applying the Energy Margin Loss, where the energy values for both ID and OOD samples overlap significantly, making it difficult to distinguish between them. After finetuning with the Energy Margin Loss, the energy values for ID samples shift toward lower energy levels, while OOD samples are pushed toward higher energy levels. This separation of energy distributions effectively enhances the model's ability to distinguish between ID and OOD samples, demonstrating the efficacy of the Energy Margin Loss in improving model robustness.

