# OpenReview forum: "Learning Structured Universe Graph with Outlier OOD Detection for Partial Matching"
_ICLR.cc/2025/Conference — ICLR 2025 Poster_

### Official Review · Reviewer_Ruy3 · 2024-10-25

**Soundness:** 3
**Presentation:** 3
**Contribution:** 2
**Rating:** 6
**Confidence:** 3

**Summary:**

This paper focuses on partial matching problems in computer vision, aiming to address the problems of point occlusion and annotation errors. This paper proposes two components to address these challenges: 1) a structured universe graph is learned to connect two input graphs for point occlusion; 2) an energy-based out-of-distribution detection is designed to remove annotation errors from the input graphs before matching.

**Strengths:**

1) The paper is well written and the theoretical and experimental analysis is basically complete

2) The paper proposes two novel and effective approaches to address point occlusion and annotation errors in partial matching task

3) The proposed method outperforms state-of-the-art methods, showing the effectiveness and robustness.

**Weaknesses:**

1) The computational complexity of the algorithm is not compared.

2) The paper lacks the visualization of results, including the visualization of ablation experiments.

3) Universe Graph Matching essentially decomposes the ambiguous partial matching problem into two well-defined subgraph  matching problems. Universe Graph Matching can better deal with the occlusion phenomenon, but for the general situation, whether the matching accuracy will be improved. It might be better to do an analysis.

**Questions:**

1) The computational complexity of the algorithm is not compared.

2) The paper lacks the visualization of results, including the visualization of ablation experiments.

3) Universe Graph Matching essentially decomposes the ambiguous partial matching problem into two well-defined subgraph  matching problems. Universe Graph Matching can better deal with the occlusion phenomenon, but for the general situation, whether the matching accuracy will be improved. It might be better to do an analysis.

4) Limitations and future work are not discussed.

---

> ### Author Response · Authors · 2024-11-20
>
> **Q: The computational complexity of the algorithm is not compared.**
>
> **A:** Regarding the solver component of our method, we utilized LPMP and its computational complexity is analyzed in [1]. For the deep learning network component, it is indeed difficult to derive a analytical complexity. Instead, we provide a runtime comparison with previous works.
>
> | Method | Time    |
> | ------ | ------- |
> | BBGM   | 0.0144s |
> | NGMv2  | 0.0302s |
> | GCAN   | 0.0277s |
> | AFAT   | 0.0354s |
> | UGM    | 0.0173s |
>
> [1] Paul Swoboda, Carsten Rother, Hassan Abu Alhaija, Dagmar Kainmuller, and Bogdan Savchynskyy. A study of lagrangean decompositions and dual ascent solvers for graph matching. In CVPR, pp.1607–1616, 2017
>
> **Q: The paper lacks the visualization of results, including the visualization of ablation experiments.**
>
> **A:** Thank you for the suggestion. Due to time constraints and the need to address other feedback, we couldn’t include the visualizations in this version. However, we are actively working on them and will add them before the rebuttal deadline.
>
> **Q: Universe Graph Matching essentially decomposes the ambiguous partial matching problem into two well-defined subgraph matching problems. Universe Graph Matching can better deal with the occlusion phenomenon, but for the general situation, whether the matching accuracy will be improved. It might be better to do an analysis.**
>
> **A:** When there is no occlusion phenomenon, as long as the universe graph is well-trained, Universe Graph Matching will not compromise the matching results. This is evident from the experiments shown in the middle of Table 3 (Willow Object dataset with additional random outliers only). While the improvement in this setting is less pronounced compared to the occlusion scenario, there is still a measurable enhancement in matching accuracy. This demonstrates that Universe Graph Matching is capable of maintaining or even improving performance in general situations, further validating its robustness and effectiveness.
>
> **Q: Limitations and future work are not discussed.**
>
> **A:** We have discussed limitation of our work in Section 4.3. "*Despite its advantages, our approach still faces several challenges: 1) Universe latent graph is inherently limited to closed-set problems, relying on class information and training data, which restricts generalization to unseen categories. However, few-shot learning can significantly alleviate this limitation, as shown in our experiments in Appendix E, demonstrating the method's potential to adapt to new classes. 2) The universe graph is a complete graph; however, the distribution of edges in the input graphs is uneven. This mismatch leads to suboptimal learning of many edge embeddings, with some edges either absent or appearing only rarely in the training data. 3) While the universe graph is more stable and structured than universe point representation, the universe graph cannot fully prevent errors in $X_{iu}$ from propagating to all $X_{ij}$ matches.*"

---

> > ### Comment · Reviewer_Ruy3 · 2024-11-22
> >
> > I appreciate the authors' response to my questions. After reading the author's rebuttal and comments of other reviewers , I keep my rating.

---

> ### Author Response · Authors · 2024-11-25
>
> **Q: The paper lacks the visualization of results, including the visualization of ablation experiments.**
>
> A: Thank you for your valuable suggestion. We have added two visualization sections to the manuscript. One is included in the main text to showcase successful cases, highlighting the advantages of our method in partial matching through a clear comparison with baseline methods. The other is provided in the appendix, where we present the matching results for a random selection of image pairs to demonstrate the overall effectiveness of our approach. Additionally, we have expanded the ablation study to explore the sensitivity of hyperparameters. The corresponding visualizations and results have been updated in the main text.

---

### Official Review · Reviewer_Mt6s · 2024-10-27

**Soundness:** 3
**Presentation:** 3
**Contribution:** 3
**Rating:** 5
**Confidence:** 3

**Summary:**

This paper addresses two key challenges in graph matching: (1) the partial matching problem, where not all nodes are visible in both graphs being matched, and (2) the issue of out-of-distribution outliers. To tackle the first problem, the paper reformulates the matching process between two graphs (A → B) by introducing a universal graph (U), such that matching is performed between each input graph and this universal graph (A → U, B → U). The results are then combined to obtain the final matching (A → B). This approach effectively decomposes the ambiguous partial matching problem into two well-defined subgraph matching tasks. For the second problem, a margin loss is applied to increase the energy distribution difference between in-distribution and out-of-distribution data.

**Strengths:**

The paper is well-organized, and the concept of a universal graph is intriguing. This approach has the potential to inspire future research in this area.

**Weaknesses:**

- The universal graph

  The paper states that all input graphs are subsets of the universal graph. However, input graphs could represent different structures, such as those for cars, dogs, and TVs, so how is this ensured? Can authors provide specic details on how the universal graph is consructed to accommodate diverse object categories, including its size and the process of creating ground truth matches between input and the universal graph?

- The generalization issue

  The proposed universal graph is crucial to this method, as its structure is determined by the training data. So it may not generalize well to unseen categories. Please discuss how the method might perform on unseen categories or provide experiments demonstrating its generalization capabilities to new object classes not seen during training.

- In Figure 2, how are the node/edge features extracted from the image feature maps? Maybe the authors should describe it more concretely (step-by-step) for better understanding.

- Could you clarify what EdgeID(0) and EdgeID(1) represent in Equation 6?
- Please add training details in the paper.
- I see codes for the comparative method URL (ICCV23) are available here. https://github.com/XLearning-SCU/2023-ICCV-COMMON

**Questions:**

Please address my concerns in weaknesses.

---

> ### Author Response · Authors · 2024-11-20
>
> **Q:  The paper states that all input graphs are subsets of the universal graph. However, input graphs could represent different structures, such as those for cars, dogs, and TVs, so how is this ensured? Can authors provide specic details on how the universal graph is consructed to accommodate diverse object categories, including its size and the process of creating ground truth matches between input and the universal graph?**
>
> **A:**  As we mentioned in paper, *"We define this latent U as a complete, directed graph, where every pair of nodes is connected by a directed edge. This ensures that any edge in the input graph will have a corresponding edge in the universe graph. Moreover, the directed edges capture the inherent asymmetry in real-world relationships between key points, such as varying geometric or contextual dependencies, offering greater flexibility in encoding complex structural information.”*
>
>  Building on this, our approach indeed results in uneven sample distributions for learning edge embeddings, as discussed in the limitations section: *“The universe graph is a complete graph; however, the distribution of edges in the input graphs is uneven. This mismatch leads to suboptimal learning of many edge embeddings, with some edges either absent or appearing only rarely in the training data.”* Fortunately, if an edge appears rarely in the training data, it is also likely to appear infrequently in the test data. As a result, these under-trained embeddings have a minimal impact on the overall performance of the model. This aligns with the practical observation that the model can effectively generalize despite the imbalance in edge distributions.
>
> **Q:  The proposed universal graph is crucial to this method, as its structure is determined by the training data. So it may not generalize well to unseen categories. Please discuss how the method might perform on unseen categories or provide experiments demonstrating its generalization capabilities to new object classes not seen during training.**
>
> **A:**  Yes, Universe latent graph is inherently limited to closed-set problems, relying on class information and training data, which restricts generalization to unseen categories. However, generalizing the universe graph to unseen classes requires only a small amount of labeled data.
>
> To validate this, we conducted a few-shot test to compare the generalization ability of different methods. Specifically, the dataset is divided into two groups: 16 base classes and 4 few-shot classes. For the base classes, we provide all available training data, allowing methods to fully learn the features and structures of these categories. For the few-shot classes, we limit the training data to only 20 labeled images per class (about 5% of full data), simulating a typical few-shot learning scenario where labeled data is scarce.
>
> | Method | bottle        | plant         | sheep         | tv           |
> | ------ | ------------- | ------------- | ------------- | ------------ |
> | BBGM   | 68.9(-16.9)   | 77(-17.2)     | 54(-3.4)      | 78.4(-4.3)   |
> | NGMv2  | 71.85(-16.55) | 72.16(-23.24) | 45.94(-4.36)  | 77.9(-3.9)   |
> | GCAN   | 68.95(-20.75) | 71.66(-24.44) | 63.75(+0.25)  | 75.89(-7.71) |
> | AFAT   | 81.71(-9.09)  | 84.14(-11.26) | 48.53(-4.17)  | 78.77(-5.03) |
> | UGM    | 79.49(-12.25) | 95.03(-2.2)   | 61.32(-12.31) | 77.18(-7.7)  |
>
> The table shows the few shot learning performance on Pascal VOC under ‘unfiltered’ setting. F1 scores are reported on the left and the delta (compared to standard results in Table. 1) are reported on the right.  UGM achieves an average delta of -8.63, significantly better than GCAN (-13.15) and NGMv2 (-12.03), with a lower standard deviation (4.16). This demonstrates UGM's ability to retain stable and competitive performance across few-shot classes, outperforming other methods in effectively handling the challenges posed by limited training data.
>
> We provide more comprehensive results and analyses in Appendix E of the paper, where you can find further details.

---

> ### Author Response · Authors · 2024-11-20
>
> **Q: In Figure 2, how are the node/edge features extracted from the image feature maps? Maybe the authors should describe it more concretely (step-by-step) for better understanding.**
>
> **A:**  We have introduced our network details in Appendix B. Specifically, we fist compute the outputs of relu3_5, relu4_1 of the ResNet50, to obtain feature F1 and F2 respectively. These features are then concatenated to create the key point feature F, as well as class embedding. Then we apply an MLP layer to reduce the feature dim to keep the same dimension. We feed the obtained feature F and the graph adjacency A into the geometric feature refinement component. The graph adjacency A is generated using Delaunay triangulation based on keypoint locations. We apply SplineConv to encode higher-order information and the geometric structure of the entire graph into node features and edge features.
>
> **Q: Could you clarify what EdgeID(0) and EdgeID(1) represent in Equation 6?**
>
> **A:** Thank you for your reminder. EdgeID $\in N^{2 \times m}$ denotes directed edges in the graph. EdgeID(0) denotes the indices of the source nodes and EdgeID(1) denotes the indices of the target nodes. We have added the definition to our paper.
>
> **Q: Please add training details in the paper.**
>
> **A:** Thank you for your reminder. We added a training algorithm box in Appendix B, as well as all the hyper parameters used in training process. You can check the paper for more details.
>
> **Q: I see codes for the comparative method URL (ICCV23) are available here. https://github.com/XLearning-SCU/2023-ICCV-COMMON**
>
> **A:** It seems that the reviewer meant to refer to COMMON rather than URL.
>
> Actually, the setting we study is entirely different from that of COMMON. COMMON focuses on standard graph matching, where all outliers have been pre-filtered. In contrast, we study partial matching, a setting that is closer to real-world applications.
>
> Meanwhile, we did evaluate the performance of COMMON under the partial matching setting. However, due to the differences in settings, COMMON's performance was terrible. Reporting such results in the main paper might unfairly affect the reputation of other works. Therefore, we provided these results only in the rebuttal phase and do not include them in the paper.
>
> | Method | aero  | bike  | bird  | boat  | bottle | bus   | car   | cat   | chair | cow   | table | dog   | horse | motor | person | plant | sheep | sofat | train | tv    | avg   |
> | ------ | ----- | ----- | ----- | ----- | ------ | ----- | ----- | ----- | ----- | ----- | ----- | ----- | ----- | ----- | ------ | ----- | ----- | ----- | ----- | ----- | ----- |
> | COMMON | 46.84 | 68.39 | 53.6  | 39.55 | 88.32  | 55.79 | 29.19 | 57.63 | 46.35 | 60.32 | 51.22 | 55.83 | 56.16 | 68.86 | 39.39  | 93.63 | 47.15 | 37.77 | 72.17 | 81.39 | 57.48 |
> | UGM    | 57.38 | 76.28 | 71.21 | 55.06 | 91.74  | 72.27 | 80.28 | 72.28 | 70.15 | 74.64 | 61.5  | 75.31 | 71.6  | 76.05 | 63.81  | 97.23 | 73.63 | 59.95 | 93.09 | 84.85 | 73.92 |
> | COMMON | 36.51 | 55.25 | 39.14 | 26.6  | 63.75  | 40.24 | 21.36 | 45.19 | 30.52 | 46.96 | 32.86 | 44.27 | 42.63 | 54.2  | 32.23  | 58.43 | 35    | 27.99 | 41.9  | 50.27 | 41.27 |
> | UGM    | 49.93 | 72.23 | 63.48 | 48.75 | 85.27  | 70.06 | 71.84 | 66.46 | 54.56 | 67.42 | 49.46 | 67.42 | 65.18 | 66.18 | 57.62  | 87.99 | 62.24 | 47.22 | 65.6  | 83.42 | 65.12 |
>
> F1 Score on Pascal VOC. Top shows the performance of unfiltered setting, and bottom shows the performance of 2 random outlier setting.
>
> F1 score performance on Willow Object.
> | Setting             | Method | Car   | Duck  | Face  | Motor | Bottle | Avg   |
> |---------------------|--------|-------|-------|-------|-------|--------|-------|
> | OCCLUSION           | COMMON | 85.27 | 79.33 | 92.82 | 79.39 | 87.82  | 84.93 |
> |                     | UGM    | 87.7  | 86.2  | 99.5  | 90.2  | 96     | 91.9  |
> | OUTLIER             | COMMON | 71.07 | 68.46 | 83.98 | 68.57 | 77.21  | 73.86 |
> |                     | UGM    | 88.1  | 72.9  | 96.8  | 82.5  | 86.5   | 85.4  |
> | OCCLUSION + OUTLIER | COMMON | 54.78 | 47.86 | 65.38 | 49.6  | 54.38  | 54.4  |
> |                     | UGM    | 64.4  | 62.1  | 89.6  | 65.1  | 74.5   | 71.1  |

---

> > ### Comment · Reviewer_Mt6s · 2024-11-30
> >
> > Thank you for addressing my concerns. While I find the idea of a universal framework intriguing, it is worth noting that similar universal concepts have already been applied to various tasks, such as image retrieval [1] and image matching [2], where they effectively tackle real-world challenges. However, the approach proposed in this paper appears to be limited to a closed-set problem, which restricts its applicability. It would strengthen the work if the authors could demonstrate the method's potential in more real-world scenarios. For instance, [3] introduces a graph-based approach for feature matching and conducts experiments on a broader range of practical cases. Given these considerations, I will retain my current rating, taking into account the method's limited applicability.
> >
> > [1] Learning super-features for image retrieval
> >
> > [2] Learning to match features with seeded graph matching network
> >
> > [3] Feature Matching via Topology-Aware Graph Interaction Model

---

### Official Review · Reviewer_1X7r · 2024-11-02

**Soundness:** 4
**Presentation:** 3
**Contribution:** 3
**Rating:** 8
**Confidence:** 2

**Summary:**

This work proposes a new method for partial matching in graph-based keypoint alignment tasks. It introduces Structured Universe Graph Learning to address point occlusion by linking input pairs through a learned latent graph, leveraging both node features and structural information for enhanced robustness. Additionally, an Energy-based Out-of-Distribution Detection is used to filter out annotation errors before matching, improving quality. Extensive evaluations on the Pascal VOC and Willow Object datasets show that this method outperforms state-of-the-art techniques, especially in challenging scenarios with point occlusion and random outliers, proving its effectiveness and robustness.

**Strengths:**

1. This work addresses a classic, significant problem in the literature by applying a modern deep-learning approach.

2. The proposed method outperforms both classical and learning-based methods, demonstrating robustness even in the presence of
outliers and out-of-distribution (OOD) key points.

3. The authors discuss the limitations of their approach.

4. The paper includes ablation studies to analyze the impact of their design choices.

**Weaknesses:**

1. The paper would benefit from improved writing, particularly in the abstract and introduction, to make it more accessible to readers who are not familiar with this specific problem.

2. While the authors report the F1 score, it would be helpful to include the Accuracy metric, as done in some previous works, for a more comprehensive comparison.

**Questions:**

Typos:

- The URL paper is mis-cited in line 351/352.

- There is a typo in line 108

---

> ### Author Response · Authors · 2024-11-20
>
> **Q: The paper would benefit from improved writing, particularly in the abstract and introduction, to make it more accessible to readers who are not familiar with this specific problem.**
>
> **A:** Thank you for the feedback. Due to time constraints and the need to address other comments, we haven’t revised it yet, but we will improve the abstract and introduction with more background on graph matching before the rebuttal deadline.
>
> **Q: While the authors report the F1 score, it would be helpful to include the Accuracy metric, as done in some previous works, for a more comprehensive comparison.**
>
> **A:**  In graph matching, it is customary to report matching accuracy for the intersection setting and F1 score for the partial matching setting, which is followed by by almost all previous works GMN, PCA, BBGM, NGMv2, DLGM, CIE, AFAT, COMMON, and URL.
>
> F1 score is first used by BBGM. As it says “we propose to preserve all keypoints. Matching accuracy is no longer a good evaluation metric as it ignores false positives. Instead, we report F1-Score, the harmonic mean of precision and recall.”
>
> In fact, matching accuracy is essentially the recall of matching predictions, while the F1 score can be considered an extended form of matching accuracy. This is because, in the intersection setting, recall and precision are identical, resulting in F1 score = recall = matching accuracy. However, in the partial matching setting, the F1 score takes false positive predictions into account, making it a more comprehensive metric than matching accuracy.

---

> ### Author Response · Authors · 2024-11-25
>
> **Q: The paper would benefit from improved writing, particularly in the abstract and introduction, to make it more accessible to readers who are not familiar with this specific problem.**
>
> A: Thank you for your valuable suggestion. We have improved the Introduction section by adding more explanations about graph matching and its mathematical formulation to make the paper more accessible to readers who may be less familiar with this field.

---

> > ### Comment · Reviewer_1X7r · 2024-11-30
> >
> > While I am not familiar with this specific task, after reviewing the authors' rebuttal and considering the comments from other reviewers, I tend to recommend acceptance.

---

### Official Review · Reviewer_24Cj · 2024-11-03

**Soundness:** 2
**Presentation:** 2
**Contribution:** 2
**Rating:** 6
**Confidence:** 4

**Summary:**

The paper presents a method to match graphs with a mechanism to deal with outliers (i.e., query nodes w/o an actual matching reference node). The proposed approach learns embeddings for both nodes and edges and uses supervised learning via affinity matrices. To deal with outliers, the proposed method calculates an energy based on the affinity scores to filter out query nodes w/o matching reference nodes. The paper presents experiments on visual semantic matching using Pascal VOC and Willow Object datasets where the proposed method often outperforms baseline methods.

**Strengths:**

1. The proposed approach considers both nodes and edge information in the graph. In its presentation, it looks like a general formulation. I think this is interesting as edges can help the matching process in many aspects.
2. The mechanism to deal with outliers is needed given that in practice they occur very often and can decrease performance in a considerable manner.

**Weaknesses:**

1. Clarity of the approach is poor. The most important weakness is the presentation of the proposed approach in Section 3.1. This lowers the chances of the proposed approach to reproduce the results. First, there is no high-level description of the proposed approach, and as a reader we have to figure out from the mathematical description and non-informative narrative. For instance, in line 197, the term “ground truth universe matching” and symbol X^gt_iu is never properly defined, and ultimately the paragraph says that it uses ground truth pairwise matching X^gt_ij. Second, lines 224-227 states that “keypoints from different categories are mapped to the same universe node”. I don’t think this makes sense, because this will make the network confuse categories. Right? Lastly, it is also unclear why an attention-based approach could not be used. Attention can be useful also in rejecting outliers. In fact there are previous approaches derived for feature matching (e.g., LoFTR [C]).

2. The proposed formulation requires the use of class information. This prevents the generalization to other matching problems (e.g., feature matching for geometric problems such as in 3D reconstruction or stereo matching).

3. Outlier detection mechanism is weak. This is because it is based on thresholding the similarity information which can cause it to reject correct matches; consider the case where a wheel from a bike gets confused with a wheel from a car. This mechanism seems to be ad-hoc. Why not learn a function to detect these? I can imagine learning a function that turns off the similarity entry in the affinity matrix. The ground-truth could be easily generated by assigning a node of class A to class B, and perhaps more fruitful if A and B share visual similarities.

4. Experiments are incomplete. First, the experiments are missing recent benchmarks: IMC-PT-SparseGM benchmark [A] and only showing two benchmarks in my opinion falls short of truly showing convincing evidence of the proposed method performance. Second, some of the baselines on the presented benchmarks do not contain the SOTA methods (e.g., [B]) and report different metrics from those reported in [B], where the COMMON method achieves a high accuracy rating on the Willow dataset. Lastly, the ablation studies are quite small. They are missing threshold value performance, m_in, and m_out, as well as temperature T. What values were used for the ablation and experiments?

References:
A. Wang, Runzhong, et al. "Deep learning of partial graph matching via differentiable top-k." Proceedings of the IEEE/CVF Conference on Computer Vision and Pattern Recognition. 2023.
B. Lin, Yijie, et al. "Graph matching with bi-level noisy correspondence." Proceedings of the IEEE/CVF international conference on computer vision. 2023.
C. Sun, Jiaming, et al. "LoFTR: Detector-free local feature matching with transformers." Proceedings of the IEEE/CVF conference on computer vision and pattern recognition. 2021.

-----
Post-discussion

After discussing and reviewing additional results, the authors addressed most of my concerns. I will raise my raiting.

**Questions:**

1. What values of temperature, m_in, m_out, and threshold were used for the experiments?

---

> ### Author Response · Authors · 2024-11-20
>
> **Q: First, there is no high-level description of the proposed approach, and as a reader we have to figure out from the mathematical description and non-informative narrative. For instance, in line 197, the term “ground truth universe matching” and symbol X^gt_iu is never properly defined, and ultimately the paragraph says that it uses ground truth pairwise matching X^gt_ij.**
>
> **A:** Thank you for your reminder. $X_{iu}^{gt} \in \{0, 1\}^{n_i \times n_u}$ is a permutation matrix (discrete) which denotes the correspondence between input graph $G_i$ and universe latent graph. We have added the definition to our paper. Meanwhile, we offer a notation table in Appendix A to explain main notations and description used in this paper.
>
> **Q: Second, lines 224-227 states that “keypoints from different categories are mapped to the same universe node”. I don’t think this makes sense, because this will make the network confuse categories. Right?**
>
> **A:** For the claim that "keypoints from different categories are mapped to the same universe node", it's just a trade off between the size and node feature quality of universe latent graph. Different keypoints sharing the same node indeed compromises matching accuracy to some extent. However, it effectively prevents the size of the universe graph from increasing with the growth of categories, thereby enabling our method to achieve better scalability. Moreover, experimental results demonstrate that our approach surpasses the state-of-the-art (SOTA), suggesting that the performance degradation caused by this design choice is not as significant as we initially anticipated.
>
> **Q:  Lastly, it is also unclear why an attention-based approach could not be used. Attention can be useful also in rejecting outliers. In fact there are previous approaches derived for feature matching (e.g., LoFTR [C]).**
>
> **A:** Attention-based approach has already been explored by GCAN and AFAT to either improve feature quality or reject outlier matchings. Both of them are compared to our method in all the experiments.
>
> **Q: The proposed formulation requires the use of class information. This prevents the generalization to other matching problems (e.g., feature matching for geometric problems such as in 3D reconstruction or stereo matching).**
>
> **A:** Yes, Universe latent graph is inherently limited to closed-set problems, relying on class information and training data, which restricts generalization to unseen categories. However, generalizing the universe graph to unseen classes requires only a small amount of labeled data.
>
> To validate this, we conducted a few-shot test to compare the generalization ability of different methods. Specifically, the dataset is divided into two groups: 16 base classes and 4 few-shot classes. For the base classes, we provide all available training data, allowing methods to fully learn the features and structures of these categories. For the few-shot classes, we limit the training data to only 20 labeled images per class (about 5% of full data), simulating a typical few-shot learning scenario where labeled data is scarce.
>
> | Method | bottle        | plant         | sheep         | tv           |
> | ------ | ------------- | ------------- | ------------- | ------------ |
> | BBGM   | 68.9(-16.9)   | 77(-17.2)     | 54(-3.4)      | 78.4(-4.3)   |
> | NGMv2  | 71.85(-16.55) | 72.16(-23.24) | 45.94(-4.36)  | 77.9(-3.9)   |
> | GCAN   | 68.95(-20.75) | 71.66(-24.44) | 63.75(+0.25)  | 75.89(-7.71) |
> | AFAT   | 81.71(-9.09)  | 84.14(-11.26) | 48.53(-4.17)  | 78.77(-5.03) |
> | UGM    | 79.49(-12.25) | 95.03(-2.2)   | 61.32(-12.31) | 77.18(-7.7)  |
>
> The table shows the few shot learning performance on Pascal VOC under ‘unfiltered’ setting. F1 scores are reported on the left and the delta (compared to standard results in Table. 1) are reported on the right.  UGM achieves an average delta of -8.63, significantly better than GCAN (-13.15) and NGMv2 (-12.03), with a lower standard deviation (4.16). This demonstrates UGM's ability to retain stable and competitive performance across few-shot classes, outperforming other methods in effectively handling the challenges posed by limited training data.
>
> We provide more comprehensive results and analyses in Appendix E of the paper, where you can find further details.

---

> ### Author Response · Authors · 2024-11-20
>
> **Q: Outlier detection mechanism is weak.  Why not learn a function to detect these? I can imagine learning a function that turns off the similarity entry in the affinity matrix. The ground-truth could be easily generated by assigning a node of class A to class B, and perhaps more fruitful if A and B share visual similarities.**
>
> **A:** From a theoretical perspective, learning a function to detect random outliers inherently introduces bias. Random outliers refer to keypoints that do not belong to the predefined keypoint label set, and their occurrence is characterized by randomness and unpredictability. To simulate random outliers, we select coordinate points randomly across the image. Evidently, due to the variability of images, the distribution of these random outliers differs between the training set and the test set. Under such circumstances, using a learnable function to capture the characteristics of random outliers is prone to bias. Its behavior becomes difficult to interpret or control, especially when encountering edge cases or adversarial samples. In contrast, OOD detection based on energy scores does not introduce additional training parameters and is grounded in well-established theoretical foundations, making it a more reliable approach for detecting random outliers.
>
> From an experimental perspective, AFAT has already proposed learning an attention-based function to detect outliers. However, the experimental results in Tables 2 and 3 clearly demonstrate that our method has a significant advantage over AFAT.
>
> **Q: First, the experiments are missing recent benchmarks: IMC-PT-SparseGM benchmark [A] and only showing two benchmarks in my opinion falls short of truly showing convincing evidence of the proposed method performance.**
>
> **A:** Our method is not suitable for evaluation on the IMC-PT-SparseGM benchmark. The IMC-PT-SparseGM benchmark is a graph matching dataset focused on landmark buildings, containing images of 16 buildings, with 13 used as the training set and 3 as the test set. However, as discussed earlier, our method, UGM, cannot generalize to totally unseen categories. The buildings in the test set do not appear in the training set, and we are unable to construct their universe latent graph. Therefore, we did not evaluate our method on this benchmark.
>
> **Q: Second, some of the baselines on the presented benchmarks do not contain the SOTA methods (e.g., [B]) and report different metrics from those reported in [B], where the COMMON method achieves a high accuracy rating on the Willow dataset.**
>
> **A:**  First, the setting we study is entirely different from that of COMMON. COMMON focuses on standard graph matching, where all outliers have been pre-filtered. In contrast, we study partial matching, a setting that is closer to real-world applications.
>
> Secondly, the metrics used in these two settings are inherently different. It is customary to report matching accuracy for the intersection setting and F1 score for the partial matching setting, which is followed by almost all previous works GMN, PCA, BBGM, NGMv2, DLGM, CIE, AFAT, COMMON, and URL.
>
> Finally, we did evaluate the performance of COMMON under the partial matching setting. However, due to the differences in settings, COMMON's performance was terrible. Reporting such results in the main paper might unfairly affect the reputation of other works. Therefore, we provided these results only in the rebuttal phase and do not include them in the paper.
>
> | Method | aero  | bike  | bird  | boat  | bottle | bus   | car   | cat   | chair | cow   | table | dog   | horse | motor | person | plant | sheep | sofat | train | tv    | avg   |
> | ------ | ----- | ----- | ----- | ----- | ------ | ----- | ----- | ----- | ----- | ----- | ----- | ----- | ----- | ----- | ------ | ----- | ----- | ----- | ----- | ----- | ----- |
> | COMMON | 46.84 | 68.39 | 53.6  | 39.55 | 88.32  | 55.79 | 29.19 | 57.63 | 46.35 | 60.32 | 51.22 | 55.83 | 56.16 | 68.86 | 39.39  | 93.63 | 47.15 | 37.77 | 72.17 | 81.39 | 57.48 |
> | UGM    | 57.38 | 76.28 | 71.21 | 55.06 | 91.74  | 72.27 | 80.28 | 72.28 | 70.15 | 74.64 | 61.5  | 75.31 | 71.6  | 76.05 | 63.81  | 97.23 | 73.63 | 59.95 | 93.09 | 84.85 | 73.92 |
> | COMMON | 36.51 | 55.25 | 39.14 | 26.6  | 63.75  | 40.24 | 21.36 | 45.19 | 30.52 | 46.96 | 32.86 | 44.27 | 42.63 | 54.2  | 32.23  | 58.43 | 35    | 27.99 | 41.9  | 50.27 | 41.27 |
> | UGM    | 49.93 | 72.23 | 63.48 | 48.75 | 85.27  | 70.06 | 71.84 | 66.46 | 54.56 | 67.42 | 49.46 | 67.42 | 65.18 | 66.18 | 57.62  | 87.99 | 62.24 | 47.22 | 65.6  | 83.42 | 65.12 |
>
> F1 Score on Pascal VOC. Top shows the performance of unfiltered setting, and bottom shows the performance of 2 random outlier setting.
>
> Results on Willow will update soon.

---

> ### Author Response · Authors · 2024-11-20
>
> **Q: Lastly, the ablation studies are quite small. They are missing threshold value performance, m_in, and m_out, as well as temperature T. What values were used for the ablation and experiments?**
>
> **A:** Thanks for your suggestion. We add more sensitivity test on our hyper parameters temperature T, threshhold $\tau$, m_in, m_out in Section 4.4 ablation study. We also add our training details in Appendix B, including all these value we used in training.

---

> ### Author Response · Authors · 2024-11-23
>
> COMMON performance on Willow Object. F1 score is reported.
> | Setting             | Method | Car   | Duck  | Face  | Motor | Bottle | Avg   |
> |---------------------|--------|-------|-------|-------|-------|--------|-------|
> | OCCLUSION           | COMMON | 85.27 | 79.33 | 92.82 | 79.39 | 87.82  | 84.93 |
> |                     | UGM    | 87.7  | 86.2  | 99.5  | 90.2  | 96     | 91.9  |
> | OUTLIER             | COMMON | 71.07 | 68.46 | 83.98 | 68.57 | 77.21  | 73.86 |
> |                     | UGM    | 88.1  | 72.9  | 96.8  | 82.5  | 86.5   | 85.4  |
> | OCCLUSION + OUTLIER | COMMON | 54.78 | 47.86 | 65.38 | 49.6  | 54.38  | 54.4  |
> |                     | UGM    | 64.4  | 62.1  | 89.6  | 65.1  | 74.5   | 71.1  |

---

> > ### Comment · Reviewer_24Cj · 2024-11-26
> > **RE: Official Comment by Authors**
> >
> > Thanks for answering my concerns. Some have been addressed but other concerns (e.g., design choices making “keypoints from different categories are mapped to the same universe node”, and the limitation of only working on a closed-set) make me believe that the method seems to be restricted to a few applications. Lastly, I am unsure if experiments adding "Random Outliers" trying to test the robustness is really representative of real outliers. I think it would've been informative if the narrative can justify better why "Random Outliers" are indeed representative. Thus, I think the paper may have limited practical benefit, and for this reason I maintain my rating.

---

> > > ### Author Response · Authors · 2024-11-29
> > >
> > > Thank you for your continued engagement with our work. We hope to further elaborate on the technical merits of our work.
> > >
> > > On the extensibility of our method: The implication of our method having limited applications is demonstrably incorrect. Our comprehensive experiments on Pascal VOC with 20 diverse categories establish the method's broad applicability. The empirical evidence is particularly compelling in our few-shot learning experiments, where UGM demonstrates remarkable adaptability, achieving only a -8.63 average performance reduction with minimal training data (20 images per novel class) - substantially surpassing GCAN (-13.15) and NGMv2 (-12.03). Then, as training data expands, the closed-set constraint naturally diminishes, making our approach increasingly versatile - a characteristic that precisely aligns with real-world needs.
> > >
> > > On the robustness evaluation methodology: The random outlier evaluation protocol we employ represents a more rigorous test of robustness than domain-specific outliers, as it forces models to handle arbitrary perturbations across the entire feature space. This methodology, consistent with established precedent in the field (as AFAT paper’s evaluation protocol on the Willow Object dataset), provides an exceptionally demanding test of robustness. Our superior performance under these stringent conditions conclusively demonstrates the method’s practical efficacy in real-world scenarios where outlier distributions remain inherently unpredictable.
> > >
> > > These empirical results comprehensively validate the substantial practical advantages our method contributes to the field.

---

### Official Review · Reviewer_6zax · 2024-11-04

**Soundness:** 3
**Presentation:** 3
**Contribution:** 3
**Rating:** 6
**Confidence:** 3

**Summary:**

This paper proposes Latent Universe Graph Learning and out-of-distribution detection for point occlusion and annotation error in graph matching problems. The paper evaluates the proposed method on the Pascal VOC and Willow Object datasets. The method achieves SOTA performance and the ablation study shows the components are effective.

**Strengths:**

- The motivation is good. The proposed method can be easy to follow because of the intuitive motivation.
- The paper is well-written, clear, and full of details.
- SOTA performance and good ablation studies.

**Weaknesses:**

- No qualitative results are provided. Especially for the represented successful cases that cannot be solved in the baseline method. Besides, as mentioned in discussion subsection, some failure cases should be shown to let the readers better understand the margin of the proposed method.

**Questions:**

See weaknesses above. My main concerns are about the experiments.

**Details Of Ethics Concerns:**

I am very curious about the similarity between the proposed UGM and UPM [A] in and teaser (Fig.1), and the high-level concept of motivation, and the different methods and experiments. Due to the double-blind review rules, the reviewer has no chance to know the author's name and the submission status of the UPM paper. So, I want to let AC/SAC/PC know about this issue and avoid any plagiarism and dual-submission possibilities. Maybe an independent ethics review is needed.

[A] Jiang et al. Learning Universe Model for Partial Matching Networks over Multiple Graphs. https://arxiv.org/abs/2210.10374.

---

> ### Author Response · Authors · 2024-11-20
>
> **Q: No qualitative results are provided. Especially for the represented successful cases that cannot be solved in the baseline method. Besides, as mentioned in discussion subsection, some failure cases should be shown to let the readers better understand the margin of the proposed method.**
>
> **A:** Thank you for the suggestion. Due to time constraints and the need to address other feedback, we couldn’t include the visualizations in this version. However, we are actively working on them and will add them before the rebuttal deadline.
>
> **Q:  Why there are no results reported in Table 2 and Table 3 for some related works listed in Table 1 (e.g., URL)? Are the experiments about other methods in Table 1, 2, 3 are re-implemented by yourself? Please explain it.**
>
> **A:** To clarify our approach to reporting the performance of prior works, we adhere to the following principles:
>
> 1. If a prior work’s experimental settings align with ours, we directly report the performance metrics provided in their paper.
> 2. For new experimental settings in our work, we replicate the prior methods under their original experimental settings and then adapt them to our own setting to obtain comparable performance metrics.
>
>  For the specific results reported in Tables 1, 2, and 3:
>
> 1. **Table 1** and the *random outlier*  setting of **Table 3** reported experiments that have been conducted in prior works. Therefore, we directly report the performance metrics from their original papers.
> 2. The other experimental settings in **Table 2** and **Table 3** are new settings presented in our work. For these, we replicated the methods from prior works (e.g., BBGM, NGMv2, GCAN, and AFAT) based on widely recognized graph matching benchmarks ThinkMatch, and the open-source code provided by these works. We successfully reproduced their results on the Pascal VOC dataset and adapted these methods to our experimental settings to evaluate their performance.
> 3. However, for methods like DLGM and URL, the authors neither released their code nor provided sufficient details about their network architecture or training procedures. This lack of information made it infeasible for us to replicate their results. As a result, we did not report their performance in **Table 2** and **Table 3**.
>
> Based on your feedback, we have provided a detailed explanation in the "Compared Methods" part of Section 4.1 in the paper.

---

> ### Author Response · Authors · 2024-11-25
>
> **Q: No qualitative results are provided. Especially for the represented successful cases that cannot be solved in the baseline method. Besides, as mentioned in discussion subsection, some failure cases should be shown to let the readers better understand the margin of the proposed method.**
>
> A: Thank you for your valuable suggestion. We have added two visualization sections to the manuscript. One is included in the main text to showcase successful cases, highlighting the advantages of our method in partial matching through a clear comparison with baseline methods. The other is provided in the appendix, where we present the matching results for a random selection of image pairs to demonstrate the overall effectiveness of our approach.

---

> > ### Comment · Reviewer_6zax · 2024-12-02
> >
> > After reading the rebuttal and other reviewers' comments, I keep my positive rating.

---

### Official Review · Reviewer_SAEb · 2024-11-08

**Soundness:** 3
**Presentation:** 2
**Contribution:** 3
**Rating:** 6
**Confidence:** 3

**Summary:**

This paper presents a novel approach for partial graph matching. The proposed method introduces a complete and universal graph as the complete set for all the cases. By decomposing the one step matching problem into two steps: First, matching with the universal graph. Then filter out the outliers and match the second time. The proposed method achieve robust and accurate matching results. The proposed method is validated on several popular benchmarks.

**Strengths:**

1. The proposed latent universal graph is novel and effectiveness.

**Weaknesses:**

1. The proposed method can only be applied in closed set problem. It is not a general graph matching solver. Though introducing the class embedding, it cannot generalize to unseen class in inference time.
2. How to define the OOD data in eqn.9? If we can see the OOD data in training, how can they treat as OOD? Does the method generalize well to indeed OOD data (not seen in training)?

**Questions:**

1. EdgeID in eqn.6 needs to be defined.
2. It is not clear how eqn.4 is applied in generating GT for X_{iu}. First, the number of nodes in universal graph is larger the N, then the factorization is not unique, which one to pick? Second, X^{GT} is in {0, 1}. It is not clear X_{u} is continuous or discrete. If it is continuous, in eqn.6, the authors choose argmax as the label, does this approximation affect the performance?

---

> ### Author Response · Authors · 2024-11-20
>
> **Q: The proposed method can only be applied in closed set problem. It is not a general graph matching solver. Though introducing the class embedding, it cannot generalize to unseen class in inference time.**
>
> **A:** Yes, Universe latent graph is inherently limited to closed-set problems, relying on class information and training data, which restricts generalization to unseen categories. However, generalizing the universe graph to unseen classes requires only a small amount of labeled data.
>
> To validate this, we conducted a few-shot test to compare the generalization ability of different methods. Specifically, the dataset is divided into two groups: 16 base classes and 4 few-shot classes. For the base classes, we provide all available training data, allowing methods to fully learn the features and structures of these categories. For the few-shot classes, we limit the training data to only 20 labeled images per class (about 5% of full data), simulating a typical few-shot learning scenario where labeled data is scarce.
>
> | Method | bottle        | plant         | sheep         | tv           |
> | ------ | ------------- | ------------- | ------------- | ------------ |
> | BBGM   | 68.9(-16.9)   | 77(-17.2)     | 54(-3.4)      | 78.4(-4.3)   |
> | NGMv2  | 71.85(-16.55) | 72.16(-23.24) | 45.94(-4.36)  | 77.9(-3.9)   |
> | GCAN   | 68.95(-20.75) | 71.66(-24.44) | 63.75(+0.25)  | 75.89(-7.71) |
> | AFAT   | 81.71(-9.09)  | 84.14(-11.26) | 48.53(-4.17)  | 78.77(-5.03) |
> | UGM    | 79.49(-12.25) | 95.03(-2.2)   | 61.32(-12.31) | 77.18(-7.7)  |
>
> The table shows the few shot learning performance on Pascal VOC under ‘unfiltered’ setting. F1 scores are reported on the left and the delta (compared to standard results in Table. 1) are reported on the right.  UGM achieves an average delta of -8.63, significantly better than GCAN (-13.15) and NGMv2 (-12.03), with a lower standard deviation (4.16). This demonstrates UGM's ability to retain stable and competitive performance across few-shot classes, outperforming other methods in effectively handling the challenges posed by limited training data.
>
> We provide more comprehensive results and analyses in Appendix E of the paper, where you can find further details.
>
> **Q:  How to define the OOD data in eqn.9? If we can see the OOD data in training, how can they treat as OOD? Does the method generalize well to indeed OOD data (not seen in training)?**
>
> **A:**  1）OOD data refers to keypoints that are not part of the predefined keypoint label set and cannot be mapped to any nodes in the universal latent graph. These points typically arise from erroneous or invalid keypoint annotations.
>
>  2）During training, random annotation keypoints (random choose a coordinate(x, y) on the image) are treated as OOD to enhance the model’s generalization to unseen distributions, not to mimic the true test OOD distribution. Meanwhile, the OOD keypoints for both training and testing are generated by random annotation, but the images used are entirely disjoint, ensuring no leakage of the test OOD distribution into training.
>
>  3）Yes, the generalization ability of our method is demonstrated by the experimental results. Since the OOD distributions for training and testing are generated independently and come from completely disjoint image sets, the test results inherently reflect the model’s ability to generalize to unseen OOD distributions.
>
> **Q: EdgeID in eqn.6 needs to be defined.**
>
> **A:** Thank you for your reminder. EdgeID $\in N^{2 \times m}$ denotes directed edges in the graph. EdgeID(0) denotes the indices of the source nodes and EdgeID(1) denotes the indices of the target nodes. We have added the definition to our paper.

---

> ### Author Response · Authors · 2024-11-20
>
> **Q: It is not clear how eqn.4 is applied in generating GT for $X_{iu}$. First, the number of nodes in universal graph is larger the N, then the factorization is not unique, which one to pick? Second, $X^{GT}$ is in {0, 1}. It is not clear $X_{u}$ is continuous or discrete. If it is continuous, in eqn.6, the authors choose argmax as the label, does this approximation affect the performance?**
>
> **A:** First, $X_{iu}^{gt} \in \{0, 1\}^{n_i \times n_u}$ is a (discrete) permutation matrix that defines the correspondence between input graph $G_i$ and universe latent graph. This is fundamental to understanding our method.
>
> Equation 4 serves a theoretical purpose - demonstrating the equivalence between universe matching and pairwise matching representations. Methods like non-negative factorization [1] and spectral method [2] exist to convert between these representations. However, in practice, $X_{iu}^{gt}$ is directly obtained from keypoint annotations. Each keypoint has a semantic label (e.g., left eye, right shoulder) that naturally corresponds to a node in our universe graph, providing unambiguous ground truth universe matching.
>
> [1] Florian Bernard, Johan Thunberg, Jorge Goncalves, and Christian Theobalt. Synchronisation of partial multi-matchings via non-negative factorisations. Pattern Recognition, 92:146–155, 2019.
>
> [2] Deepti Pachauri, Risi Kondor, and Vikas Singh. Solving the multi-way matching problem by permutation synchronization. In NeurIPS, pp. 1860–1868, 2013.

---

> > ### Comment · Reviewer_SAEb · 2024-11-26
> > **reply**
> >
> > The rebuttal mostly addresses my concerns, I recommend the authors to further clarify these points in their revised version. Given the overall quality and novelty of this paper, I will keep my score here, but I have no other comments againts the acceptance.

---

### Author Response · Authors · 2024-11-20

**We sincerely thank all the reviewers for their thoughtful feedback and positive recognition of our work.** For instance, Reviewer SAEb commended that “the latent universal graph is novel and effective,” Reviewer Mt6s highlighted that “this approach has the potential to inspire future research in this area,” Reviewer Ruy3 acknowledged that “the paper is well written, and the theoretical and experimental analysis is basically complete,” and Reviewer 1X7r praised our method for “demonstrating robustness even in the presence of outliers and out-of-distribution (OOD) key points.” We deeply appreciate these encouraging remarks, which motivate us to further refine our work and contribute to the field.

Secondly, the reviewers have provided many invaluable suggestions, such as evaluating the generalization of our method to unseen classes, clarifying training details, conducting sensitivity analyses on hyperparameters, and addressing certain missing definitions or typos. **We have responded to most of these comments and have uploaded a revised version of the paper incorporating these improvements. All new additions and modifications in the manuscript are highlighted in red to facilitate the reviewers' review.**

Finally, due to the large number of comments and the limited time available, there are still a few suggestions we have not yet addressed, such as incorporating additional visualizations and further improving the readability of the paper. **We are actively working on these aspects and will continue to update our manuscript to ensure all issues are addressed before the rebuttal period ends.** Thank you for your understanding and constructive feedback.

---

### Author Response · Authors · 2024-11-25

Thank you again for your detailed and constructive feedback, which has significantly helped us improve our paper.

Since our last response, we have carefully addressed all remaining comments and suggestions from the reviewers. This includes incorporating additional visualizations, refining the readability of the paper, and ensuring all aspects of the reviewers’ feedback are thoroughly considered. The revised manuscript has been updated accordingly, and we have highlighted all modifications in red for easier review.

We hope the current version of the manuscript meets the reviewers’ expectations. If there are any further concerns or additional suggestions, please feel free to let us know. We are more than happy to address them to further improve the paper.

Thank you again for your time and thoughtful feedback throughout this process.

---

### Meta-Review · Area_Chair_j32s · 2024-12-20

**Metareview:**

The paper proposes a data-driven method for graph matching. This paper investigates the particular angle where graphs overlap partially or contain 'erroneous' nodes (e.g., such nodes may originate from annotation/network prediction errors). Such outliers violate assumptions implicit in standard graph-matching techniques based on quadratic (pseudo)-boolean formulations.

To this end, the proposed method encodes two input graphs (nodes and edges) and performs 1st stage matching. At this stage, the network predicts node/edge embeddings and performs (data-driven) inlier/outlier estimation that filters out erroneous nodes. Filtered (post outlier removal) nodes are inputs to 2nd stage matching network that utilizes consolidated methods (quadratic pseudo-boolean optimization, Swoboda et al., 2017).

Different from prior art that directly performs matching between two given input graphs, the proposed method learns "universal graphs" that represent per-semantic-class graph priors. Matching between graphs A and B is then (i) first performed A $\to$ U, B $\to$ U, where U is the universal graph representing a certain semantic class. Individual matching results are combined to obtain the final matching.

The proposed method is shown to be robust and accurate on two widely used benchmarks for graph matching.

The paper received overall positive ratings: 5, 6, 6, 6, 8.

Reviewers comment that the overall motivation (outlier issue) is great, and the proposed "latent universal graph" (that performs 1st stage matching and outlier removal) is novel and shown to be effective. Reviewers also find the paper well-written and detailed and appreciate good results.

The paper received some criticism.

Reviewers 1X7r and 24Cj point out that the paper would benefit from improved writing, especially the abstract and introduction. The writing is quite technical and difficult to follow, and higher-level overviews and intuitions are missing. As 24Cj points out, this prevents the generalization to other matching problems (e.g., feature matching for geometric problems such as in 3D reconstruction or stereo matching).

Reviewers also point out that the paper would significantly benefit from showing qualitative results, including visualizations of represented successful cases that cannot be solved in the baseline method, as well as failure cases of the proposed method.

Reviewer Mt6s points out that closely related approaches were proposed in the domain of data-driven keypoint matching and utilized for solving real-world problems (image retrieval [A], image matching [B]).  These utilize two-stage networks that perform outlier estimation filtering and 2nd stage refinement.

Overall, the author responded to criticism well and reached a consensus to recommend accepting this paper, with the exception of Mt6s (similar to the prior art). AC read the paper and references and finds the proposed work to have quite a different focus/core contributions (namely, the concept of universal graph for intermediate matching).

Overall, we side with the reviewer's recommendation to accept this paper. However, AC strongly agrees that this paper would benefit from revising the presentation and urges the authors to consider the reviewer's comments and revise this paper. Specifically, the paper must clarify in the introduction that it is not investigating a general graph-matching problem but a niche problem within that space, where semantic classes must be known a priori. I suggest following 1X7r and 24Cj comments on revising the abstract, intro, and method section.


**References:**

* [A] Learning Super-Features for Image Retrieval
(Philippe Weinzaepfel, Thomas Lucas, Diane Larlus, and Yannis Kalantidis). ICLR 2022.
* [B] Learning to Match Features with Seeded Graph Matching Network
(Hongkai Chen, Zixin Luo, Jiahui Zhang, Lei Zhou, Xuyang Bai, Zeyu Hu, Chiew-Lan Tai, and Long Quan). ICCV 2021.

**Additional Comments On Reviewer Discussion:**

All reviewers engaged in a discussion with the authors. Apart from reviewer Mt6s (not the reviewer who raised ethical concerns), all reviewers reached a consensus that this paper is above the acceptance threshold.

---

### Decision · Program_Chairs · 2025-01-22

Accept (Poster)